# Learning to Learn with Contrastive Meta-Objective

## Abstract

We propose a contrastive meta-objective to enable meta-learners to emulate human-like rapid learning capability through enhanced alignment and discrimination. Our proposed approach, dubbed ConML, exploits task identity as additional supervision signal for meta-training, benefiting meta-learner's fast-adaptation and task-level generalization abilities. This is achieved by contrasting the outputs of meta-learner, i.e, performing contrastive learning in the model space. Specifically, we introduce metrics to minimize the inner-task distance, i.e., the distance among models learned on varying data subsets of the same task, while maximizing the inter-task distance among models derived from distinct tasks. ConML distinguishes itself through versatility and efficiency, seamlessly integrating with episodic meta-training methods and the in-context learning of large language models (LLMs). We apply ConML to representative meta-learning algorithms spanning optimization-, metric-, and amortization-based approaches, and show that ConML can universally and significantly improve conventional meta-learning and in-context learning.

## 1 Introduction

Meta-learning [37, 42], or learning to learn, is a powerful paradigm that aims to enable a learning system to quickly adapt to new tasks. Meta-learning has been widely applied in different fields, like few-shot learning [17, 50], reinforcement learning [56, 26] and neural architecture search [16, 38]. In meta-training, a meta-leaner mimics the learning processes on many relevant tasks to gain experience about how to make adaptation. In meta-testing, the meta-trained adaptation process is performed on unseen tasks. The adaptation process is achieved by generating task-specific model by the meta-learner, which is given a set of training examples and returns a predictive model. People prefer meta-learning to equip models with human's fast learning ability, so that a good model can be achieved with a few examples [50].

The combination of two cognitive capabilities, namely, **alignment** and **discrimination**, is essential for human's fast learning ability [23, 12, 13]. A good learner possesses the alignment [27] ability to align different partial views of a certain object, which means they can integrate various aspects or perspectives of information to form a coherent understanding. On the other hand, discrimination [34] refers to the learner's capacity to distinguish between one stimulus and similar stimuli, responding appropriately only to the correct stimuli. This is a fundamental ability that allows learners to differentiate between what is relevant and what is not, ensuring that their responses are accurate and based on the correct understanding of the stimuli presented. With alignment and discrimination, learners can synthesize fragmented information to construct a complete picture of an object or concept, while also being able to discern subtle differences between distinct but similar objects or ideas. Such learners are not only efficient in processing information but also in applying their knowledge accurately in varied contexts. This dual capability is crucial for effective learning.

We expect meta-learners to emulate the above combination of alignment and discrimination capabilities to approach human's fast learning ability. By equipping a meta-learner with the ability to

Submitted to 38th Conference on Neural Information Processing Systems (NeurIPS 2024). Do not distribute.

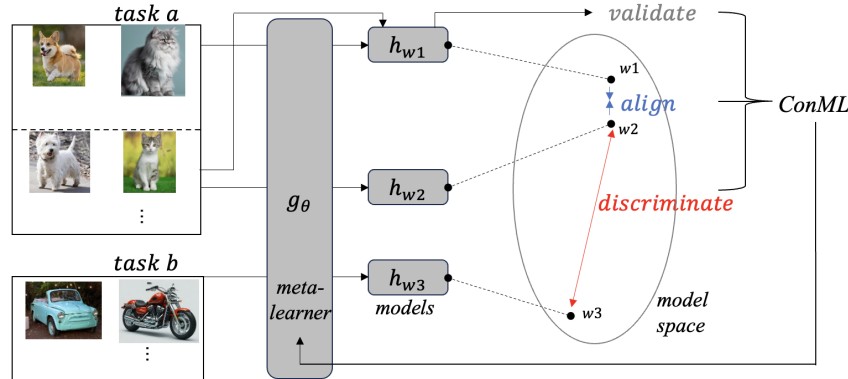

Figure 1: ConML is performing contrastive learning in model space, where alignment and discrimination encourage the meta-learner's fast-adaptation and task-level generalize ability respectively.

align, we enable it to capture the core essence of a task and being invariant to noises. Meanwhile, discrimination ensures that a meta-learner can learn specific models for unique tasks, as it is a natural supposition that different tasks enjoy distinguishable models. This reflects the natural diversity of problems we encounter in the real world and the varied strategies we employ to solve them. Together, alignment and discrimination empower a meta-learner to not only grasp the subtleties of individual tasks but also to generalize its learning across a spectrum of challenges. This dual capability can makes a meta-learner robust, versatile, and more aligned with the nuanced nature of human learning and reasoning. However, existing meta-learning approaches conventionally follows the idea of "train as you test", to minimize the validation loss [46] of meta-training tasks as meta-objective, where supervision signal are directly produced by sample labels. To provide stronger supervision, there are works assuming that the task-specific target models of meta-training tasks are available, then the meta-training can be supervised by aligning the learned model and the corresponding target model, with model weights [51, 52] or knowledge distillation [55]. However, as the target models are expensive to learn, and even not available in many real world problems, meta-objectives requiring the target models have very restricted applications. Moreover, the importance of discrimination ability of meta-learner has not been noticed in the literature.

To achieve this, we propose contrastive meta-learning (ConML), by directly contrasting the outputs of meta-learner in the model space, shown in Figure 1. Conventional contrastive learning (CL) [14, 48, 44] learns an encoder in unsupervised manner by equipping the model with alignment and discrimination ability by exploiting the distinguishable identity of unlabeled samples. Considering tasks in meta-learning are also unlabeled but have distinguishable identity, we are inspired to adopt similar strategy in meta-learning. ConML exploits tasks as CL exploits unlabeled samples. Positive pairs in ConML are different subsets of the same task, while negative pairs are datasets of different tasks. In the model space output by meta-learner, inner-task distance can be measured between positive pairs and inter-task distance can be measured between negative pairs. The contrastive meta-objective is minimizing inner-task distance while maximizing inter-task distance, corresponding to the expected alignment and discrimination ability respectively. The proposed ConML is universal and cheap, as it can be plugged-in any meta-learning algorithms following the episodic training, and does not require additional data nor model training. In this paper, we widely study ConML on representative meta-learning algorithms from different categories: optimization-based (e.g., MAML [17]), metric-based (e.g., ProtoNet [39]), amortization-based (e.g., Simple CNAPS [6]). We also investigate in-context learning [8] with reformulating it into the meta-learning paradigm, and show how ConML integrates and helps.

Our contributions are:

- We propose to emulate cognitive alignment and discrimination capabilities in meta-learning, to narrow down the gap of fast learning ability between meta-learners and humans.

- We generalize contrastive learning from representation space of unsupervised learning to model space of meta-learning. The exploiting task identity as additional supervision benefits meta-learner's fast-adaptation and task-level generalize abilities.

- ConML is algorithm-agnostic, that can be incorporated into any meta-learning algorithms with episodic training. We empirically show ConML can bring universal improvement with cheap implementation on a wide range of meta-learning algorithms and in-context learning.

## 2 Related Works

### 2.1 Learning to Learn

Meta-learning learns to improve the learning algorithm itself [37], i.e., learns to learn. Popular meta-learning approaches can be roughly divided into three categories [7]: optimization-based, metric-based and amortization-based. Optimization-based approaches [4, 17, 28] focus on learning better optimization strategies for adapting to new tasks. For example MAML [17] learns initial model parameters, where few steps of gradient descent can quickly make adaptaion for specific tasks. Metric-based approaches [46, 39, 41] leverages learned similarity metrics. For example, Prototypical Networks [39] and Matching Networks [46] learn global shared encoders to map training set to embeddings, based on which task-specific model can be built. Amortization-based approaches [19, 33, 6] seek to learn a shared representation across tasks. They amortize the adaptation process by using neural networks to directly infer task-specific parameters from training set. Examples are CNPs [19] and CNAPs [33].

In-context learning (ICL) [8] is designed for large language models, which integrates examples (input-output pairs) in a task and a query input into the prompt, thus the language model can answer the query. Recently, ICL has been studied as a general approach of learning to learn [2, 18, 47, 1], which reduces meta-learning to conventional supervised learning via training a sequence model. It considers training set as context to be provided along with the input to predict, forming a sequence to feed the model. Training such a model can be viewed as an instance of meta-learning [18].

### 2.2 Contrastive Learning

Contrastive learning is a powerful technique in representation learning [29, 10, 48]. Its primary goal is to learn useful representations, which are invariant to unnecessary details, and preserve as much information as possible. This is achieved by maximizing alignment and discrimination (uniformity) in representation space [48]. In conventional contrastive learning, alignment refers to bringing positive pairs (e.g., augmentations of the same sample [54, 22, 5, 21, 10]) closer together in the learned representation space. By maximizing alignment, the representations are encouraged to be invariant to unneeded noise factors. Discrimination refers to separating negative pairs (e.g., different samples) farther. Maximizing discrimination without any other knowledge results in uniformity, i.e., uniform distribution in the representation space. By maximizing discrimination, the representations are encouraged to preserve as much information of the data as possible [43, 5], benefiting the generalization ability.

## 3 Meta-Learning with Contrastive Meta-Objective

Meta-learning is a methodology considered with "learning to learn" machine learning algorithms. Define $\mathcal{L}(\mathcal{D}; h)$ as the loss obtained by evaluating model $h$ on dataset $\mathcal{D}$ with function $\ell(y, \hat{y})$ (e.g., cross entropy or mean squared loss), $g(; \theta)$ is a meta-learner that maps a dataset $\mathcal{D}$ to a model $h$, i.e, $h = g(\mathcal{D}; \theta)$. Given a distribution of tasks $p(\tau)$, where each task $\tau$ consists of a training set $\mathcal{D}_\tau^{\text{tr}} = \{(x_{\tau,i}, y_{\tau,i})\}_{i=1}^n$, and a validation set $\mathcal{D}_\tau^{\text{val}} = \{(x_{\tau,i}, y_{\tau,i})\}_{i=n+1}^m$, the goal of meta-learning is to learn $g(; \theta)$ to perform well on new task $\tau'$ sampled from $p(\tau')$, evaluated by $\mathcal{L}(\mathcal{D}_{\tau'}^{\text{val}}; g(\mathcal{D}_{\tau'}^{\text{tr}}; \theta))$.

### 3.1 A Unified View of Episodic Training

We aim to introduce "learning to align and discriminate" to universally improve the meta-learning process. The most conventional way of meta-training is taking the *validation loss* as meta-objective to optimize $\theta$:

$$\min_\theta \mathbb{E}_{\tau \sim p(\tau)} \mathcal{L}(\mathcal{D}_\tau^{\text{val}}; g(\mathcal{D}_\tau^{\text{tr}}; \theta)). \tag{1}$$

Different meta-learning algorithms tailor the function inside $g$, while sharing the same episodic meta-training to achieve (1). Shown as Algorithm 1, in each episode, $B$ tasks are sampled from $p(\tau)$ to form a batch **b**, and validation loss of each task is aggregated as the supervision signal $L_v = \frac{1}{B} \sum_{\tau \in \boldsymbol{b}} \mathcal{L}(\mathcal{D}_\tau^{\text{val}}; g(\mathcal{D}_\tau^{\text{tr}}; \theta))$ to update $\theta$. By specifying the function inside $g$, Algorithm 1 can generalize the meta-training process of different meta-learning algorithms.

---

**Algorithm 1** Mini-Batch Episodic Meta-Training (Conventional)

---

**while** Not converged **do**
    Sample a batch of tasks $\boldsymbol{b} \sim p^B(\tau)$.
    **for** All $\tau \in \boldsymbol{b}$ **do**
        Get task-specific model $h_\tau = g(\mathcal{D}_\tau^{\text{tr}}; \theta)$;
        Get validation loss $\mathcal{L}(\mathcal{D}_\tau^{\text{val}}; h_\tau)$;
    **end for**
    $L_v = \frac{1}{B} \sum_{\tau \in \boldsymbol{b}} \mathcal{L}(\mathcal{D}_\tau^{\text{val}}; g(\mathcal{D}_\tau^{\text{tr}}; \theta))$
    Update $\theta$ by $\theta \leftarrow \theta - \nabla_\theta L_v$.
**end while**

---

Table 1: Specifications of ConML.

| Category | Examples | $g(\mathcal{D};\theta)$ | $\psi(g(\mathcal{D};\theta))$ |
|---|---|---|---|
| Optimization -based | MAML[17], Reptile[28] | Update model weights $\theta - \nabla_\theta \mathcal{L}(\mathcal{D};h_\theta)$ | $\theta - \nabla_\theta \mathcal{L}(\mathcal{D};h_\theta)$ |
| Metric -based | ProtoNet[39], MatchNet[46] | Build classifier with $\{(\{f_\theta(x_i)\}_{x_i \in \mathcal{D}_j}, j)\}_{j=1}^N$ | Concatenate $[\frac{1}{|\mathcal{D}_j|}\sum_{x_i \in \mathcal{D}_j} f_\theta(x_i)]_{j=1}^N$ |
| Amortization -based | CNPs[19], CNAPs[33] | Map $\mathcal{D}$ to model weights by $H_\theta(\mathcal{D})$ | $H_\theta(\mathcal{D})$ |

Specifications of optimization-based, metric-based and amortization-based algorithms are summarized in Table 1.

We design ConML to be integrated with Algorithm 1 without specifying $g$, thus to be universally applicable for meta-learning algorithms following the episodic manner. In Section 3.2, we introduce how to measure the objective. Then in Section 3.3, we introduce specifications of ConML on a wide range of meta-learning algorithms.

## 3.2 Integration with Episodic Meta-Training

To equip meta-learners with the desired alignment and discrimination ability, we design contrastive meta-objective measured in the output space of meta-learner, i.e., the model space of $h$. Alignment is achieved by minimizing inner-task distance, which is the distance among models generated from different subsets of the same task. Discrimination is achieved by maximize the inter-task distance, which is the distance among models generated from different tasks. Here we introduce how to measure the contrastive objective and perform optimization.

**Obtaining Model Representation.** To train the meta-learner $g$, the distances $D^{\text{in}}, D^{\text{out}}$ are measured in the output space of $g$, i.e., the model space $\mathcal{H}$. A feasible way is to first represent model $h = g(\mathcal{D};\theta) \in \mathcal{H}$ as fixed length vectors $\boldsymbol{e} \in \mathbb{R}^d$, then measure by explicit distance function $\phi(\cdot,\cdot)$ (e.g., cosine distance). Note that $\mathcal{H}$ is algorithm-specific. Here we only introduce a projection $\psi : \mathcal{H} \to \mathbb{R}^d$ to obtain model representations $\boldsymbol{e} = \psi(h)$. The $\mathcal{H}$ and $\psi$ will be elucidated and specified for different meta-learning algorithms in Section 3.3.

**Obtaining Inner-Task Distance.** During meta-training, $\mathcal{D}_\tau^{\text{tr}} \cup \mathcal{D}_\tau^{\text{val}}$ contains all the available information about task $\tau$. The meta-learner is expected to learn similar model given any subset $\kappa$ of the task. Meanwhile those models from subsets are expected to be similar to the model learned from the full supervision $\mathcal{D}_\tau^{\text{tr}} \cup \mathcal{D}_\tau^{\text{val}}$. We design the following inner-task distance to minimize that encourages $g$ to learn a generalizable model even from a set containing only few or biased samples. For $\forall \kappa \subseteq \mathcal{D}_\tau^{\text{tr}} \cup \mathcal{D}_\tau^{\text{val}}$, we expect $\boldsymbol{e}_\tau^\kappa = \boldsymbol{e}_\tau^*$, where $\boldsymbol{e}_\tau^\kappa = \psi(g(\kappa;\theta))$, $\boldsymbol{e}_\tau^* = \psi(g(\mathcal{D}_\tau^{\text{tr}} \cup \mathcal{D}_\tau^{\text{val}};\theta))$. The inner-task distance $D_\tau^{\text{in}}$ of task $\tau$ is defined as:

$$D_\tau^{\text{in}} = \frac{1}{K}\sum_{k=1}^K \phi(\boldsymbol{e}_\tau^{\kappa_k}, \boldsymbol{e}_\tau^*), \ s.t., \ \boldsymbol{e}_\tau^{\kappa_k} \sim \pi_\kappa(\mathcal{D}_\tau^{\text{tr}} \cup \mathcal{D}_\tau^{\text{val}}), \tag{2}$$

where $\{\kappa_k\}_{k=1}^K$ are $K$ subsets sampled from $\mathcal{D}_\tau^{\text{tr}} \cup \mathcal{D}_\tau^{\text{val}}$ by certain sampling strategy $\pi_\kappa$. In each episode given a batch of task $\boldsymbol{b}$ containing $B$ tasks, inner-task distance is averaged by $D^{\text{in}} = \frac{1}{B}\sum_{\tau \in \boldsymbol{b}} D_\tau^{\text{in}}$.

**Obtaining Inter-Task Distance.** Since the goal of meta-learning is improving the performance on unseen tasks, it is important that the $g$ is generalizable for diverse tasks. With a natural supposition that different tasks enjoy different task-specific models, it is necessary that $g$ can learn different models from different tasks, i.e., discrimination. We define the following inter-task distance to maximize to improve the task-level generalizability of $g$. For two tasks $\tau \neq \tau'$ during meta-training, we expect to maximize the distance between $\boldsymbol{e}_\tau^*$ and $\boldsymbol{e}_{\tau'}^*$. To be practical under the mini-batch episodic training paradigm, we consider to measure inter-task distance among a batch of tasks:

$$D^{\text{out}} = \frac{1}{B(B-1)}\sum_{\tau \in \boldsymbol{b}}\sum_{\tau' \in \boldsymbol{b}\setminus\tau} \phi(\boldsymbol{e}_\tau^*, \boldsymbol{e}_{\tau'}^*). \tag{3}$$

**Training Procedure.** ConML measures $D^{\text{in}}$ by (2) and $D^{\text{out}}$ by (3) in each episode, and minimizes a combination of the validation loss $L_v$ and contrastive meta-objective $D^{\text{in}} - D^{\text{out}}$:

$$L = L_v + \lambda(D^{\text{in}} - D^{\text{out}}). \quad (4)$$

The training procedure of ConML is provided in Algorithm 2. Comparing with Algorithm 1, ConML introduces additional computation $\psi(g(\mathcal{D};\theta))$ for $K+1$ times in each episode. Note that we implement $\psi$ with very cheap function such as obtaining model weights (or a single probing, i.e., feeding-forward, for ICL), and $g(\mathcal{D};\theta)$ already exists in Algorithm 1 while multiple $g(\mathcal{D};\theta)$ can be parallel-computed. ConML could have very comparable time consumption.

---

**Algorithm 2** Meta-Learning with Contrastive Meta-Object (ConML)

---
**while** Not converged **do**
  Sample a batch of tasks $\boldsymbol{b} \sim p^B(\tau)$.
  **for** All $\tau \in \boldsymbol{b}$ **do**
    **for** $k = 1, 2, \cdots, K$ **do**
      Sample $\kappa_k$ from $\pi_\kappa(\mathcal{D}_\tau^{\text{tr}} \cup \mathcal{D}_\tau^{\text{val}})$;
      Get model representation $\boldsymbol{e}_\tau^{\kappa_k} = \psi(g(\kappa_k; \theta))$;
    **end for**
    Get model representation $\boldsymbol{e}_\tau^* = \psi(g(\mathcal{D}_\tau^{\text{tr}} \cup \mathcal{D}_\tau^{\text{val}}; \theta))$;
    Get inner-task distance $D_\tau^{\text{in}}$ by (2);
    Get task-specific model $h_\tau = g(\mathcal{D}_\tau^{\text{tr}}; \theta)$;
    Get validation loss $\mathcal{L}(\mathcal{D}_\tau^{\text{val}}; h_\tau)$;
  **end for**
  Get $D^{\text{in}} = \frac{1}{B} \sum_{\tau \in \boldsymbol{b}} D_\tau^{\text{in}}$ and $D^{\text{out}}$ by (3);
  Get loss $L$ by (4);
  Update $\theta$ by $\theta \leftarrow \theta - \nabla_\theta L$.
**end while**

---

### 3.3 Instantiations of ConML

Here we demonstrate specifications of $\mathcal{H}$ and $\psi(g(\mathcal{D}, \theta))$ to obtain model representation to implement ConML. We show examples on representative meta-learning algorithms from different categories: optimization-based, metric-based and amortization-based. They are explicitly represented by model weights, summarized in Table 1.

**With Optimization-Based Methods.** The representative algorithm of optimization-based meta-learning is MAML. It meta-learns an initialization from where gradient steps are taken to learn task-specific models, i.e., $g(\mathcal{D}; \theta) = h_{\theta - \nabla_\theta \mathcal{L}(\mathcal{D}; h_\theta)}$. As $g$ directly generates the model weights, we explicitly take the model weights as model representation. The representation of model learned by $g$ given a dataset $\mathcal{D}$ is $\psi(g(\mathcal{D}; \theta)) = \theta - \nabla_\theta \mathcal{L}(\mathcal{D}; h_\theta)$. Note that there are optimization-based meta-learning algorithms which are based on first-order approximation of MAML, thus they do not strictly follows Algorithm 1 to minimize validation loss (e.g., FOMAML [17] and Reptile [28]). ConML can also be incorporated as long as it follows the episodic manner.

**With Metric-Based Methods.** Metric-based algorithms are feasible for classification tasks. Given dataset $\mathcal{D}$ of a $N$-way classification task, metric-based algorithms can be summarized as classifying according to distances with $\{\{f_\theta(x_i)\}_{x_i \in \mathcal{D}_j}\}_{j=1}^N$ and corresponding labels, where $f_\theta$ is a meta-learned encoder and $\mathcal{D}_j$ is the set of inputs belongs to class $j$. We design to represent this metric-based classifier with the concatenation of mean embedding of each class in label-aware order. For example, ProtoNet [39] computes the prototype $\boldsymbol{c}_j$, i.e., mean embedding of samples in each class. $\boldsymbol{c}_j = \frac{1}{|\mathcal{D}_j|} \sum_{(x_i, y_i) \in \mathcal{D}_j} f_\theta(x_i)$. Then classifier $h_{\theta, \mathcal{D}}$ is built by giving prediction $p(y = j \mid x) = \exp(-d(f_\theta(x), \boldsymbol{c}_j)) / \sum_{j'} \exp(-d(f_\theta(x), \boldsymbol{c}_{j'}))$. As the outcome model $h_{\theta, \mathcal{D}}$ depends on $\mathcal{D}$ through $\{\boldsymbol{c}_j\}_{j=1}^N$ and corresponding labels, the representation is specified as $\psi(g(\mathcal{D}; \theta)) = [\boldsymbol{c}_1 | \boldsymbol{c}_2 | \cdots | \boldsymbol{c}_N]$, where $[\cdot | \cdot]$ means concatenation.

**With Amortization-Based Methods.** Amortization-based approaches meta-learns a hypernetwork $H_\theta$, which aggregates information from $\mathcal{D}$ to task-specific parameter $\alpha$ and serves as weights of main-network $h$, resulting in task-specific model $h_\alpha$. For example, Simple CNAPS [6] adopts the hypernetwork to generate only a small amount of task-specific parameter, which performs feature-wise linear modulation (FiLM) on convolution channels of the main-network. For contrasting we represent $h_\alpha$ by $\alpha$, i.e., the output of hypernetwork $H_\theta$: $\psi(g(\mathcal{D}; \theta)) = H_\theta(\mathcal{D})$. The detailed procedures of different meta-learning algorithms with ConML are provided in Appendix A.

## 4  In-Context Learning with Contrastive Meta-Objective

In-context learning (ICL) is first proposed for large language models [8], where examples in a task are integrated into the prompt (input-output pairs) and given a new query input, the language model can generate the corresponding output. This approach allows pre-trained model to address new tasks without fine-tuning the model. For example, given "*happy->positive; sad->negative; blue->*", the model can output "*negative*", while given "*green->cool; yellow->warm; blue->*" the model can output "*cool*". ICL has the ability to learn from the prompt. Training ICL can be viewed as learning

to learn, like meta-learning [25, 18, 24]. More generally, the input and output are not necessarily to be natural language. In ICL, a sequence model $T_\theta$ (typically transformer [45]) is trained to map sequence $[x_1, y_1, x_2, y_2, \cdots, x_{m-1}, y_{m-1}, x_m]$ (prompt prefix) to prediction $y_m$. Given distribution $P$ of training prompt $t$, then training ICL follows an auto-regressive manner:

$$\min_\theta \mathbb{E}_{t\sim P(t)} \frac{1}{m} \sum_{i=0}^{m-1} \ell(y_{t,i+1}, T_\theta([x_{t,1}, y_{t,1}, \cdots, x_{t,i+1}])). \tag{5}$$

It has been mentioned that the training of ICL can be viewed as an instance of meta-learning [18, 2] as $T_\theta$ learns to learn from prompt. In this section we first formally reformulate $T_\theta$ to meta-learner $g(; \theta)$, then introduce how ConML can be integrated with ICL.

### 4.1 A Meta-learning Reformulation

Denote a sequentialized $\mathcal{D}$ as $\vec{\mathcal{D}}$ where the sequentializer is default to bridge $p(\tau)$ and $P(t)$. Then the prompt $[x_{\tau,1}, y_{\tau,1}, \cdots, x_{\tau,m}, y_{\tau,m}]$ can be viewed as $\vec{\mathcal{D}}_\tau^{tr}$ which is providing task-specific information. Note that ICL does not specify an explicit output model $h(x) = g(\mathcal{D}; \theta)(x)$; instead, this procedure exists only implicitly through the feeding-forward of the sequence model, i.e., task-specific prediction is given by $g([\vec{\mathcal{D}}, x]; \theta)$. Thus we can reformulate the training of ICL (5) as:

$$\min_\theta \mathbb{E}_{\tau\sim p(\tau)} \frac{1}{m} \sum_{i=0}^{m-1} \ell(y_{\tau,i+1}, g([\vec{\mathcal{D}}_{\tau,0:i}, x_{\tau,i+1}]; \theta)). \tag{6}$$

Equation (6) can be regarded as the validation loss (1) in meta-learning, where each task in each episode is sampled multiple times to form $\mathcal{D}_\tau^{\text{val}}$ and $\mathcal{D}_\tau^{\text{tr}}$ in an auto-regressive manner. The training of ICL thus follows the episodic meta-training (Algorithm 1), where the validation loss with determined $\mathcal{D}_\tau^{\text{tr}}$ and $\mathcal{D}_\tau^{\text{val}}$: $\mathcal{L}(\mathcal{D}_\tau^{\text{val}}; g(\mathcal{D}_\tau^{\text{tr}}; \theta))$, is replaced by loss validated in the auto-regressive manner: $\frac{1}{m} \sum_{i=0}^{m-1} \ell(y_{\tau,i+1}, g([\vec{\mathcal{D}}_{\tau,0:i}, x_{\tau,i+1}]; \theta))$.

### 4.2 Integration with ICL

Since the training of ICL could be reformulated as episodic meta-training, the three steps to measure ConML proposed in Section 3.2 can be also adopted for ICL, but the first step to obtain model representation $\psi(g(\mathcal{D}, \theta))$ needs modification. Due to the absence of an inner learning procedure for a predictive model for prediction $h(x) = g(\mathcal{D}; \theta)(x)$, representation by explicit model weights of $h$ is not feasible for ICL.

To represent what $g$ learns from $\mathcal{D}$, we design to incorporate $\vec{\mathcal{D}}$ with a dummy input $u$, which functions as a probe and its corresponding output can be readout as representation:

$$\psi(g(\mathcal{D}; \theta)) = g([\vec{\mathcal{D}}, u]; \theta), \tag{7}$$

where $u$ is constrained to be in the same shape as $x$, and has consistent value in an episode. The complete algorithm of ConML for ICL is provided in Appendix A. From the perspective of learning to learn, ConML encourages ICL to align and discriminate like it does for conventional meta-learning, while the representations to evaluate inner- and inter- task distance are obtained by probing output rather than explicit model weights. Thus, incorporating ConML into the training process of ICL benefits the fast-adaptation and task-level generalization ability. From the perspective of supervised learning, ConML is performing unsupervised data augmentation that it introduces the dummy input and contrastive objective as additional supervision to train ICL.

## 5 Experiments

In this secrion, we first empirically investigate the alignment and discrimination empowered by ConML. Then we show the effect of ConML that it significantly improve meta-learning performance on a wide range of meta-learning algorithms on few-shot image classification, and the effect of ConML-ICL with in-context learning general functions. Additionally, by applying ConML we provide a SOTA approach for few-shot molecular property prediction problem, provided in Appendix B. Code is provided in supplementary materials.

### 5.1 Impact of Alignment and Discrimination

There are two important questions to understand the way ConML works: First, does ConML equip meta-learners with better alignment and discrimination as expected? Second, what is the contribution of inner-task and inter-task distance respectively? We take ConML-MAML as example and investigate above questions with few-shot regression problem following the same settings in [17], where each task involves regressing from the input to the output of a sine wave. We use this synthetic regression

Table 2: Meta-testing and clustering performance of few-shot sinusoidal regression.

| Method | MSE (5-shot) | MSE (10-shot) | Silhouette | DBI | CHI |
|---|---|---|---|---|---|
| MAML | $.6771 \pm .0377$ | $.0678 \pm .0022$ | $.1068 \pm .0596$ | $.0678 \pm .0021$ | $31.55 \pm 2.52$ |
| ConML-MAML | $\mathbf{.3935} \pm .0100$ | $\mathbf{.0397} \pm .0009$ | $\mathbf{.1945} \pm .0621$ | $\mathbf{.0397} \pm .0009$ | $\mathbf{39.22} \pm 2.61$ |

dataset to be able to sample data and vary the distribution as needed for investigation. The implement of ConML-MAML is consistent with Section 5.2. Firstly the meta-testing performance in Table 2 shows that ConML is effective for the regression problem.

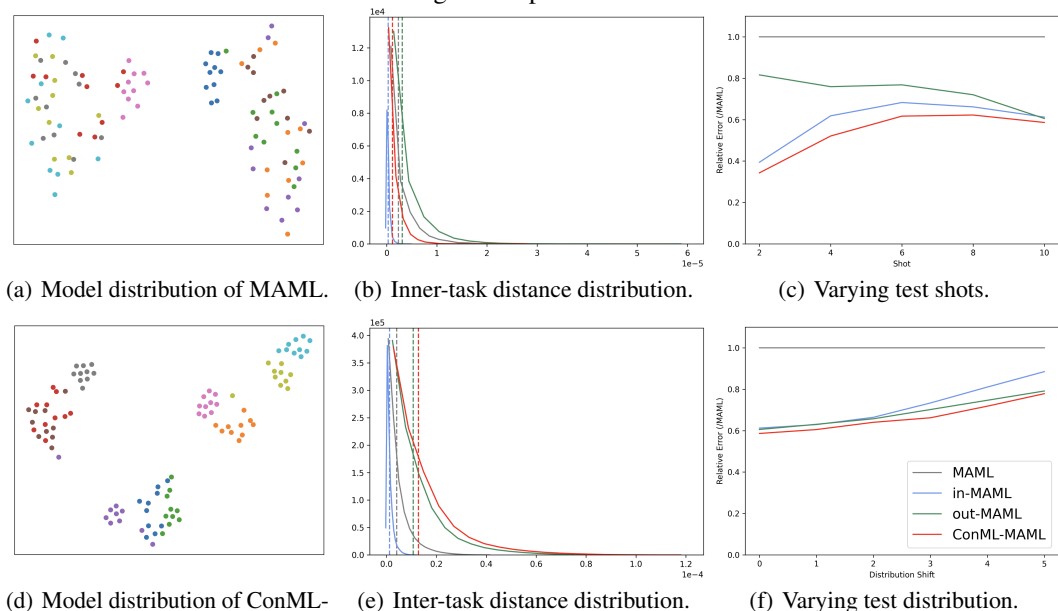

(a) Model distribution of MAML.  (b) Inner-task distance distribution.  (c) Varying test shots.

(d) Model distribution of ConML-MAML.  (e) Inter-task distance distribution.  (f) Varying test distribution.

Figure 2: Investigating the way ConML works.

**Clustering.** If ConML enhances the alignment and discrimination abilities, ConML-MAML can generate more similar models from different subsets of the same task, while generating more separable models from different tasks. This can be verified by evaluating the clustering performance for model representations $e$. During meta-testing, we randomly sample 10 different tasks, inside each we sample 10 different subsets, each one contains $N = 10$ samples. Taking these 100 different $\mathcal{D}^{\mathrm{tr}}$ as input, meta-learner generates 100 models. Figure 2(a) and 2(d) show the visualization of model distribution. It can be obviously observed ConML-MAML performs better alignment and discrimination than MAML. To quantity the results, we also evaluate the supervised clustering performance, where task identity is used as label. Table 2 shows the supervised clustering performance of different metrics: Silhouette score [35], Davies-Bouldin index (DBI) [15] and Calinski-Harabasz index (CHI) [9], where ConML-MAML shows much better performance.

**Decoupling Inner- and Inter-Task Distance.** In conventional unsupervised contrastive learning, where objective only relies on contrasting of positive pairs and negative pairs, positive and negative pairs are both necessary to avoid learning representations without useful information. However, in ConML, there is validation loss $L_v$ plays a necessary and fundamental role in "learning to learn", and the contrastive objective is introduced as additional supervision to enhance alignment and discrimination. Thus, distance of positive pairs ($D^{\mathrm{in}}$) and negative pairs ($D^{\mathrm{out}}$) in ConML could be decoupled and incorporated with $L_v$ respectively. We aim to understand how $D^{\mathrm{in}}$ and $D^{\mathrm{out}}$ contributes respectively. This gives birth to two variants of ConML: **in-MAML** which optimize $L_v$ and $D^{\mathrm{in}}$, **out-MAML** which optimize $L_v$ and $D^{\mathrm{out}}$. During meta-testing, we randomly sample 1000 different tasks, inside each we sample 10 different subsets each one contains $N = 10$ samples. We aggregate different subsets from the same task to form a $N = 100$ set to obtaining $e_\tau^*$ for each task. The distribution of $D^{\mathrm{in}}$ and $D^{\mathrm{out}}$ are shown in Figure 2(b) and 2(e) respectively, where the dashed lines are mean values. We can find that: the alignment and discrimination ability corresponds to optimizing $D^{\mathrm{in}}$ and $D^{\mathrm{out}}$ respectively; the alignment and discrimination capabilities are generalizable; ConML shows the couple of both capabilities. Figure 2(c) shows the testing performance given different numbers of examples per task (shot), while the meta-leaner is trained with fixed $N = 10$. We can find that the improvement brought by $D^{\mathrm{in}}$ is much more significant than $D^{\mathrm{out}}$ under few-shot scenario, which indicates that alignment is closely related to the fast-adaptation ability of the meta-learner.

Table 3: Meta-testing accuracy on *mini*ImageNet.

| Category | Algorithm | Setting (5-way) | w/o ConML | ConML- | Relative Gain | Relative Time |
|---|---|---|---|---|---|---|
| Optimization-Based | MAML | 1-shot
5-shot | $48.75 \pm 1.25$
$64.50 \pm 1.02$ | $\mathbf{56.25 \pm 0.94}$
$\mathbf{67.37 \pm 0.97}$ | 9.16% | 1.1× |
| | FOMAML | 1-shot
5-shot | $48.12 \pm 1.40$
$63.86 \pm 0.95$ | $\mathbf{57.64 \pm 1.29}$
$\mathbf{68.50 \pm 0.78}$ | 12.65% | 1.2× |
| | Reptile | 1-shot
5-shot | $49.21 \pm 0.60$
$64.31 \pm 0.97$ | $\mathbf{52.82 \pm 1.06}$
$\mathbf{67.04 \pm 0.81}$ | 5.58% | 1.5× |
| Metric-Based | MatchNet | 1-shot
5-shot | $43.92 \pm 1.03$
$56.26 \pm 0.90$ | $\mathbf{48.75 \pm 0.88}$
$\mathbf{62.04 \pm 0.89}$ | 10.59% | 1.2× |
| | ProtoNet | 1-shot
5-shot | $48.90 \pm 0.84$
$65.69 \pm 0.96$ | $\mathbf{51.03 \pm 0.91}$
$\mathbf{67.35 \pm 0.72}$ | 3.31% | 1.2× |
| Amortization-Based | SCNAPs | 1-shot
5-shot | $53.14 \pm 0.88$
$70.43 \pm 0.76$ | $\mathbf{55.73 \pm 0.86}$
$\mathbf{71.70 \pm 0.71}$ | 3.12% | 1.3× |

Figure 2(f) shows the out-of-distribution testing performance. While meta-trained on tasks with amplitudes that uniformly distribute on $[0.1, 5]$, meta-testing is performed on tasks with amplitudes that uniformly distribute on $[0.1 + \delta, 5 + \delta]$ (the distribution shift $\delta$ is indicated as $x$-axis). We can find that the improvement brought by $D^{\text{out}}$ is notably more significant as the distribution gap grows than $D^{\text{in}}$. This indicates that discrimination is closely related to the task-level generalization ability of meta-learner. ConML takes both advantages brought by $D^{\text{in}}$ and $D^{\text{out}}$.

## 5.2 Few-Shot Image Classification

To evaluate ConML on conventional meta-learning approaches, we follow existing works [46, 17, 39, 28, 6] to evaluate the meta-learning performance with few-shot image classification problem. We consider representative meta-learning algorithms from different categories, including optimization-based: **MAML** [17], **FOMAML** [17], **Reptile** [28]; metric-based: **MatchNet** [46], **ProtoNet** [39]; and amortization-based: **SCNAPs** (Simple CNAPS) [6]. We evaluate their original meta-learning performance (**w/o ConML**) and performance meta-trained with the proposed ConML (**ConML-**). The implementation of ConML- follows the general Algorithm 2 and the specification for corresponding category in Section 3.3.

**Datasets and Settings.** We consider two few-shot image classification benchmarks: *mini*ImageNet [46] and *tiered*ImageNet [32]. 5-way 1-shot and 5-way 5-shot tasks are trained and evaluated respectively. Note that we focus on the improvement comparing ConML- and the corresponding algorithm without ConML, rather than performance comparison across different algorithms. So we conduct the experiment on each algorithm following the originally reported settings. All baselines share the same settings of hyperparameters related to the measurement of ConML: task batch size $B = 32$, inner-task sampling $K = 1$ and $\pi_\kappa(\mathcal{D}_\tau^{\text{tr}} \cup \mathcal{D}_\tau^{\text{val}}) = \mathcal{D}_\tau^{\text{tr}}$, $\phi(a, b) = 1 - a \cdot b / \|a\|\|b\|$ (cosine distance) and $\lambda = 0.1$. For other settings of hyperparameters about model architecture and training procedure, each baseline is consistent with its originally reported. Note that $K = 1$ and $\pi_\kappa(\mathcal{D}_\tau^{\text{tr}} \cup \mathcal{D}_\tau^{\text{val}}) = \mathcal{D}_\tau^{\text{tr}}$ is the most simple and efficient implementation, provided as *Efficient*-ConML in Appendix A. In this case, considering the consumption of feeding-forward neural networks in each task, Algorithm 1 takes $h = g(\mathcal{D}_\tau^{\text{tr}}; \theta)$ and $\mathcal{L}(\mathcal{D}_\tau^{\text{val}}; h)$, while ConML only introduces an additional $g(\mathcal{D}_\tau^{\text{tr}} \cup \mathcal{D}_\tau^{\text{val}}; \theta)$, which results in very comparable time consumption.

**Results.** Table 3 and 4 show the results on *mini*ImageNet and *tiered*ImageNet respectively. The relative gain is calculated in terms of the summation of 1-shot and 5-shot accuracy. The relative time is comparing the total time consumption of meta-training. Significant relative gain and very comparable relative time consumption show that ConML brings universal improvement on different meta-learning algorithms with cheap implementation.

## 5.3 In-Context Learning General Functions

Following [18], we investigate ConML on ICL by learning to learn synthetic functions including linear regression (LR), sparse linear regression (SLR), decision tree (DT) and 2-layer neural network with ReLU activation (NN). We train the GPT-2 [30]-like transformer for each function with ICL and ConML-ICL respectively and compare the inference (meta-testing) performance. We follow the same model structure, data generation and training settings [18]. We implement ConML-ICL with $K = 1$ and $\pi_\kappa([x_1, y_1, \cdots, x_n, y_n]) = [x_1, y_1, \cdots, x_{\lfloor \frac{n}{2} \rfloor}, y_{\lfloor \frac{n}{2} \rfloor}]$. To obtain the implicit representation (7), we sample $u$ from a standard normal distribution (the same with $x$'s distribution) independently in

Table 4: Meta-testing accuracy on *tiered*ImageNet.

| Category | Algorithm | Setting (5-way) | w/o ConML | ConML- | Relative Gain | Relative Time |
|---|---|---|---|---|---|---|
| Optimization-Based | MAML | 1-shot | $51.39 \pm 1.31$ | $\mathbf{58.75} \pm 1.45$ | 10.07% | 1.1× |
| | | 5-shot | $68.25 \pm 0.98$ | $\mathbf{72.94} \pm 0.98$ | | |
| | FOMAML | 1-shot | $51.44 \pm 1.51$ | $\mathbf{58.21} \pm 1.22$ | 9.78% | 1.2× |
| | | 5-shot | $68.32 \pm 0.95$ | $\mathbf{73.26} \pm 0.78$ | | |
| | Reptile | 1-shot | $47.88 \pm 1.62$ | $\mathbf{55.01} \pm 1.28$ | 10.78% | 1.5× |
| | | 5-shot | $65.10 \pm 1.13$ | $\mathbf{70.15} \pm 1.00$ | | |
| Metric-Based | MatchNet | 1-shot | $48.74 \pm 1.06$ | $\mathbf{53.29} \pm 1.05$ | 11.00% | 1.2× |
| | | 5-shot | $61.30 \pm 0.94$ | $\mathbf{67.86} \pm 0.77$ | | |
| | ProtoNet | 1-shot | $52.50 \pm 0.96$ | $\mathbf{54.62} \pm 0.79$ | 3.94% | 1.2× |
| | | 5-shot | $71.03 \pm 0.74$ | $\mathbf{73.78} \pm 0.75$ | | |
| Amortization-Based | SCNAPs | 1-shot | $62.88 \pm 1.04$ | $\mathbf{65.06} \pm 0.95$ | 2.91% | 1.3× |
| | | 5-shot | $79.82 \pm 0.87$ | $\mathbf{81.79} \pm 0.80$ | | |

Table 5: Performance comparison of ConML-ICL and ICL.

| Function (max prompt len.) | LR (10 shot) | SLR (10 shot) | DT (20 shot) | NN (40 shot) |
|---|---|---|---|---|
| Rel. Min. Error | $0.42 \pm 0.09$ | $0.49 \pm .06$ | $0.81 \pm 0.12$ | $0.74 \pm 0.19$ |
| Shot Spare | $-4.68 \pm 0.45$ | $-3.94 \pm 0.62$ | $-4.22 \pm 1.29$ | $-11.25 \pm 2.07$ |

each episode. Since the output of (7) is a scalar, i.e., representation $e \in \mathbb{R}$, we adopt distance measure $\phi(a, b) = \sigma((a - b)^2)$, where $\sigma(\cdot)$ is sigmoid function to bound the squared error. $\lambda = 0.02$.

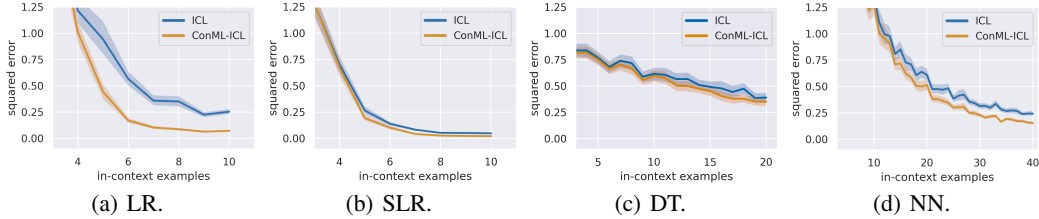

(a) LR.     (b) SLR.     (c) DT.     (d) NN.

Figure 3: In-context learning performance.

**Results.** Figure 3 shows that varying the number of in-context examples during inference, ConML-ICL always makes more accurate predictions than ICL. Table 5 collects the two values to show the effect ConML brings to ICL: *Rel. Min. Error* is ConML-ICL's minimal inference error given different number of examples, divided by ICL's; *Shot Spare* is when ConML-ICL obtain an error no larger than ICL's minimal error, the difference between the corresponding example numbers. Note that the learning of different functions (different meta-datasets) share the same settings about ConML, which shows ConML can bring ICL universal improvement with cheap implementation. We notice that during training of LR and SLR $\lfloor \frac{n}{2} \rfloor = 5$, which happens to equals to the dimension of the regression task. This means sampling by $\pi_\kappa$ would results in the minimal sufficient information to learn the task. In this case, minimizing $D^{\text{in}}$ is particularly beneficial for the fast-adaptation ability, shown as Figure 3(a) and 3(b). This indicates that introducing prior knowledge to design the hyperparameter settings of ConML could bring more advantage. The effect of ConML for ICL is without loss of generalizability to real-world applications like pretraining large language models.

## 6 Conclusion

In this work, we propose ConML that introduce an additional supervision for episodic meta-training by exploiting task identity. The contrastive meta-objective is designed to emulate the alignment and discrimination embodied in human's fast learning ability, and measured by performing contrastive learning in the model space. Specifically, we design ConML to be integrated with the conventional episodic meta-training, and then give specifications on a wide range of meta-learning algorithms. We also reformulate training ICL into episodic meta-training to design ConML-ICL following the same principle. Empirical results show that ConML can universally and significantly improve meta-learning performance by benefiting the meta-learner's fast-adaptation and task-level generalization ability. This work lays the groundwork for contrastive meta-learning, by identifying the importance of alignment and discrimination ability of meta-learner, and practicing contrastive learning in model space. There also exists certain limitations, such as lack of investigating advanced contrastive strategy, batch- and subset- sampling strategies. We would consider these as future directions.

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

 # A  Specifications of ConML

---

**Algorithm 3** ConML

---

**Input:** Task distribution $p(\tau)$, batch size $B$, inner-task sample times $K$ and sampling strategy $\pi_\kappa$.
**while** Not converged **do**
   Sample a batch of tasks $\boldsymbol{b} \sim p^B(\tau)$.
   **for** All $\tau \in \boldsymbol{b}$ **do**
     **for** $k = 1, 2, \cdots, K$ **do**
       Sample $\kappa_k$ from $\pi_\kappa(\mathcal{D}_\tau^{\text{tr}} \cup \mathcal{D}_\tau^{\text{val}})$;
       Get model representation $\boldsymbol{e}_\tau^{\kappa_k} = \psi(g(\kappa_k; \theta))$;
     **end for**
     Get model representation $\boldsymbol{e}_\tau^* = \psi(g(\mathcal{D}_\tau^{\text{tr}} \cup \mathcal{D}_\tau^{\text{val}}; \theta))$;
     Get inner-task distance $D_\tau^{\text{in}}$ by (2);
     Get task-specific model $h_\tau = g(\mathcal{D}_\tau^{\text{tr}}; \theta)$;
     Get validation loss $\mathcal{L}(\mathcal{D}_\tau^{\text{val}}; h_\tau)$;
   **end for**
   Get $D^{\text{in}} = \frac{1}{B} \sum_{\tau \in \boldsymbol{b}} D_\tau^{\text{in}}$ and $D^{\text{out}}$ by (3);
   Get loss $L$ by (4);
   Update $\theta$ by $\theta \leftarrow \theta - \nabla_\theta L$.
**end while**

---

---

**Algorithm 4** *Efficient* ConML

---

**Input:** Task distribution $p(\tau)$, batch size $B$ (inner-task sample times $K = 1$ and sampling strategy $\pi_\kappa(\mathcal{D}_\tau^{\text{tr}} \cup \mathcal{D}_\tau^{\text{val}}) = \mathcal{D}_\tau^{\text{tr}}$).
**while** Not converged **do**
   Sample a batch of tasks $\boldsymbol{b} \sim p^B(\tau)$.
   **for** All $\tau \in \boldsymbol{b}$ **do**
     Get task-specific model $h_\tau = g(\mathcal{D}_\tau^{\text{tr}}; \theta)$, and model representation $\boldsymbol{e}_\tau^{\kappa_k} = \psi(g(\kappa_k; \theta))$;
     Get model representation $\boldsymbol{e}_\tau^* = \psi(g(\mathcal{D}_\tau^{\text{tr}} \cup \mathcal{D}_\tau^{\text{val}}; \theta))$;
     Get inner-task distance $D_\tau^{\text{in}}$ by (2);
     Get validation loss $\mathcal{L}(\mathcal{D}_\tau^{\text{val}}; h_\tau)$;
   **end for**
   Get $D^{\text{in}} = \frac{1}{B} \sum_{\tau \in \boldsymbol{b}} D_\tau^{\text{in}}$ and $D^{\text{out}}$ by (3);
   Get loss $L$ by (4);
   Update $\theta$ by $\theta \leftarrow \theta - \nabla_\theta L$.
**end while**

---

---

**Algorithm 5** In-Context Learning with Contrastive Meta-Object (ConML-ICL)

---

**Input:** Task distribution $p(\tau)$, batch size $B$, inner-task sample times $K$ and sampling strategy $\pi_\kappa$, dummy input $u$ (probe).
**while** Not converged **do**
    Sample a batch of tasks $\boldsymbol{b} \sim p^B(\tau)$.
    **for** All $\tau \in \boldsymbol{b}$ **do**
        **for** $k = 1, 2, \cdots, K$ **do**
            Sample $\kappa_k$ from $\pi_\kappa(\mathcal{D}_\tau)$;
            Get $\boldsymbol{e}_\tau^{\kappa_k} = g([\vec{\kappa_k}, u]; \theta)$;
        **end for**
        Get $\boldsymbol{e}_\tau^* = g([\vec{\mathcal{D}}_\tau, u]; \theta)$;
        Get inner-task distance $D_\tau^{\text{in}}$ by (2);
        Get task loss $\frac{1}{m} \sum_{i=0}^{m-1} \ell(y_{\tau, i+1}, g([\vec{\mathcal{D}}_{\tau, 0:i}, x_{\tau, i+1}]; \theta))$;
    **end for**
    Get $D^{\text{in}} = \frac{1}{B} \sum_{\tau \in \boldsymbol{b}} D_\tau^{\text{in}}$ and $D^{\text{out}}$ by (3);
    Get loss $L = \frac{1}{B} \sum_{\tau \in \boldsymbol{b}} \frac{1}{m} \sum_{i=0}^{m-1} \ell(y_{\tau, i+1}, g([\vec{\mathcal{D}}_{\tau, 0:i}, x_{\tau, i+1}]; \theta)) + \lambda(D^{\text{in}} - D^{\text{out}})$;
    Update $\theta$ by $\theta \leftarrow \theta - \nabla_\theta L$.
**end while**

---

**Algorithm 6** ConML-MAML

---

**Input:** Task distribution $p(\tau)$, batch size $B$, inner-task sample times $K = 1$ and sampling strategy $\pi_\kappa$
**while** Not converged **do**
    Sample a batch of tasks $\boldsymbol{b} \sim p^B(\tau)$.
    **for** All $\tau \in \boldsymbol{b}$ **do**
        **for** $k = 1, 2, \cdots, K$ **do**
            Sample $\kappa_k$ from $\pi_\kappa(\mathcal{D}_\tau^{\text{tr}} \cup \mathcal{D}_\tau^{\text{val}})$;
            Get model representation $\boldsymbol{e}_\tau^{\kappa_k} = \theta - \nabla_\theta \mathcal{L}(\kappa_k; h_\theta)$;
        **end for**
        Get model representation $\boldsymbol{e}_\tau^* = \theta - \nabla_\theta \mathcal{L}(\mathcal{D}_\tau^{\text{tr}} \cup \mathcal{D}_\tau^{\text{val}}; h_\theta)$.
        Get inner-task distance $D_\tau^{\text{in}}$ by (2);
        Get task-specific model $h_{\theta - \nabla_\theta \mathcal{L}(\mathcal{D}_\tau^{\text{tr}}; \theta)}$;
        Get validation loss $\mathcal{L}(\mathcal{D}_\tau^{\text{val}}; h_{\theta - \nabla_\theta \mathcal{L}(\mathcal{D}_\tau^{\text{tr}}; h_\theta)})$;
    **end for**
    Get $D^{\text{in}} = \frac{1}{B} \sum_{\tau \in \boldsymbol{b}} D_\tau^{\text{in}}$ and $D^{\text{out}}$ by (3);
    Get loss $L$ by (4);
    Update $\theta$ by $\theta \leftarrow \theta - \nabla_\theta L$.
**end while**

---

**Algorithm 7** ConML-Reptile

---

**Input:** Task distribution $p(\tau)$, batch size $B$. (inner-task sample times $K = 1$ and sampling strategy $\pi_\kappa(\mathcal{D}_\tau^{\text{tr}} \cup \mathcal{D}_\tau^{\text{val}}) = \mathcal{D}_\tau^{\text{tr}}$)
**while** Not converged **do**
    Sample a batch of tasks $\boldsymbol{b} \sim p^B(\tau)$.
    **for** All $\tau \in \boldsymbol{b}$ **do**
        **for** $k = 1, 2, \cdots, K$ **do**
            Sample $\kappa_k$ from $\pi_\kappa(\mathcal{D}_\tau)$;
            Get model representation $\boldsymbol{e}_\tau^{\kappa_k} = \theta - \nabla_\theta \mathcal{L}(\kappa_k; h_\theta)$;
        **end for**
        Get model representation $\boldsymbol{e}_\tau^* = \theta - \nabla_\theta \mathcal{L}(\mathcal{D}_\tau^{\text{tr}} \cup \mathcal{D}_\tau^{\text{val}}; h_\theta)$.
        Get inner-task distance $D_\tau^{\text{in}}$ by (2);
    **end for**
    Get $D^{\text{in}} = \frac{1}{B} \sum_{\tau \in \boldsymbol{b}} D_\tau^{\text{in}}$ and $D^{\text{out}}$ by (3);
    Get loss $L$ by (4);
    Update $\theta$ by $\theta \leftarrow \theta + \frac{1}{B} \sum_{\tau \in \boldsymbol{b}} (\boldsymbol{e}_\tau^* - \theta) - \lambda \nabla_\theta (D^{\text{in}} - D^{\text{out}})$.
**end while**

---

---

**Algorithm 8** ConML on SCNAPs

---

**Note:** Here $h_w$ corresponds to the feature extractor $f_\theta$; $H_\theta$ corresponds to the task encoder $g_\phi$ in [6].
**Input:** Task distribution $p(\tau)$, batch size $B$, inner-task sample times $K$ and sampling strategy $\pi_\kappa$.
Pretrain $h_w$ with the mixture of all meta-training data;
**while** Not converged **do**
    Sample a batch of tasks $\boldsymbol{b} \sim p^B(\tau)$.
    **for** All $\tau \in \boldsymbol{b}$ **do**
        **for** $k = 1, 2, \cdots, K$ **do**
            Sample $\kappa_k$ from $\pi_\kappa(\mathcal{D}_\tau^{\mathrm{tr}} \cup \mathcal{D}_\tau^{\mathrm{val}})$;
            Get model representation $\boldsymbol{e}_\tau^{\kappa_k} = H_\theta(\kappa_k)$;
        **end for**
        Get model representation $\boldsymbol{e}_\tau^* = H_\theta(\mathcal{D}_\tau^{\mathrm{tr}} \cup \mathcal{D}_\tau^{\mathrm{val}})$;
        Get inner-task distance $D_\tau^{\mathrm{in}}$ by (2);
        Get task-specific model by FiLM $h_\tau = h_{w, H_\theta(\mathcal{D}_\tau^{\mathrm{tr}})}$;
        Get validation loss $\mathcal{L}(\mathcal{D}_\tau^{\mathrm{val}}; h_\tau)$;
    **end for**
    Get $D^{\mathrm{in}} = \frac{1}{B} \sum_{\tau \in \boldsymbol{b}} D_\tau^{\mathrm{in}}$ and $D^{\mathrm{out}}$ by (3);
    Get loss $L$ by (4);
    Update $\theta$ by $\theta \leftarrow \theta - \nabla_\theta L$.
**end while**

---

---

**Algorithm 9** ConML-ProtoNet ($N$-way classification)

---

**Input:** Task distribution $p(\tau)$, batch size $B$, inner-task sample times $K = 1$ and sampling strategy $\pi_\kappa$
**while** Not converged **do**
    Sample a batch of tasks $\boldsymbol{b} \sim p^B(\tau)$.
    **for** All $\tau \in \boldsymbol{b}$ **do**
        **for** $k = 1, 2, \cdots, K$ **do**
            Sample $\kappa_k$ from $\pi_\kappa(\mathcal{D}_\tau^{\mathrm{tr}} \cup \mathcal{D}_\tau^{\mathrm{val}})$;
            Calculate prototypes $\boldsymbol{c}_j = \frac{1}{|\kappa_{k,j}|} \sum_{(x_i, y_i) \in \kappa_{k,j}} f_\theta(x_i)$ for $j = 1, \cdots, N$;
            Get model representation $\boldsymbol{e}_\tau^{\kappa_k} = [\boldsymbol{c}_1 | \boldsymbol{c}_2 | \cdots | \boldsymbol{c}_N]$;
        **end for**
        Calculate prototypes $\boldsymbol{c}_j = \frac{1}{|\mathcal{D}_j|} \sum_{(x_i, y_i) \in \mathcal{D}_j} f_\theta(x_i)$ for $j = 1, \cdots, N$;
        Get model representation $\boldsymbol{e}_\tau^* = [\boldsymbol{c}_1 | \boldsymbol{c}_2 | \cdots | \boldsymbol{c}_N]$;
        Get inner-task distance $D_\tau^{\mathrm{in}}$ by (2);
        Get task-specific model $h_{[\boldsymbol{c}_1 | \boldsymbol{c}_2 | \cdots | \boldsymbol{c}_N]}$, which gives prediction by $p(y = j \mid x) = \frac{exp(-d(f_\theta(x), \boldsymbol{c}_j))}{\sum_{j'} exp(-d(f_\theta(x), \boldsymbol{c}_{j'}))}$;
        Get validation loss $\mathcal{L}(\mathcal{D}_\tau^{\mathrm{val}}; h_{[\boldsymbol{c}_1 | \boldsymbol{c}_2 | \cdots | \boldsymbol{c}_N]})$;
    **end for**
    Get $D^{\mathrm{in}} = \frac{1}{B} \sum_{\tau \in \boldsymbol{b}} D_\tau^{\mathrm{in}}$ and $D^{\mathrm{out}}$ by (3);
    Get loss $L$ by (4);
    Update $\theta$ by $\theta \leftarrow \theta - \nabla_\theta L$.
**end while**

---

## B  Few-shot Molecular Property Prediction

Few-shot molecular property prediction (FSMPP) is an important real-world application where meta-learning has been widely applied recently [3, 20, 49, 11, 36]. Molecular property prediction, which predicts whether desired properties will be active on given molecules, plays a crucial role in many applications like computational chemistry [31] and drug discovery [53]. As wet-lab experiments to evaluate the actual properties of molecules are expensive and risky, usually only a few labeled molecules are available for a specific property. Molecular property prediction can be naturally modeled as a few-shot learning problem [3], and meta-learning approaches has been successfully adopted for FSMPP [3, 20, 49, 11].

**Dataset and Settings.**  We use FS-Mol [40], a widely studied FSMPP benchmark consisting of a large number of diverse tasks. We adopt the public data split [40]. Each training set contains 64 labeled molecules, and can be imbalanced where the number of labeled molecules from active and inactive is not equal. All remaining molecules in the task form the validation set. The performance is evaluated by $\Delta$AUPRC (change in area under the precision-recall curve) w.r.t. a random classifier [40], averaged across meta-testing tasks.

**Baselines.**  We consider the following meta-learning-based FSMPP approaches: **MAML**, **ProtoNet**, **CNP**, **IterRefLSTM**, **PAR**, **ADKF-IFT**. Note that MHNfs [36] is not included as it uses additional reference molecules from external datasets, which leads to unfair comparison, and ADKF-IFT is the SOTA approach in literature. All baselines share the same GNN-based encoder provided by the benchmark to meta-train from scratch, which maps molecular graphs to embedding vectors.

---

**Algorithm 10** Hypro

**Note:** The main-network consists of two modules [40]: the molecular encoder $f_\theta$ and the prototypical network classifier $h_\theta$.
**Input:** Task distribution $p(\tau)$, batch size $B$.
**while** Not converged **do**
  Sample a batch of tasks $\boldsymbol{b} \sim p^B(\tau)$.
  **for** All $\tau \in \boldsymbol{b}$ **do**
    Encode all molecules $f_\theta(x)$ for $x \in \mathcal{D}_\tau^{\text{tr}} \cup \mathcal{D}_\tau^{\text{val}}$
    Get task-specific parameters $\alpha_\tau = H_\theta(\{(f_\theta(x_i), y_i)\}_{(x_i,y_i)\in\mathcal{D}_\tau^{\text{tr}}})$;
    Modulate all molecular embedding with $\alpha_\tau$ by FiLM, and classify with $h_\theta$; (denote the function of this step as $h_{\theta,\alpha_\tau}$)
    Get validation loss $\mathcal{L}(\mathcal{D}_\tau^{\text{val}}; h_{\theta,\alpha_\tau})$;
  **end for**
  $L_v = \frac{1}{B}\sum_{\tau\in\boldsymbol{b}} \mathcal{L}(\mathcal{D}_\tau^{\text{val}}; h_{\theta,\alpha_\tau})$
  Update $\theta$ by $\theta \leftarrow \theta - \nabla_\theta L_v$.
**end while**

---

We introduce a new baseline **ConML-Hypro**, which achieves SOTA performance by incorporating ConML with a simple backbone, **Hypro**. It is an amortization-based model built by modifying the ProtoNet backbone, by plugging-in a hypernetwork $H$ with a set-encoder structure, i.e., $H(\mathcal{D}) = \text{MLP}_2\left(\frac{1}{|\mathcal{D}|}\sum_\mathcal{D} \text{MLP}_1([x_i \mid y_i])\right)$. We input the embedding vectors in $\mathcal{D}^{\text{tr}}$ to the hypernetwork, and take the output to modulate embedding vectors through FiLM before classification. This hypernetwork and modulation is typical in amortization-based models. Viewing Hypro as an amortization-based model, we apply the specification of ConML to form ConML-Hypro. The detailed procedure to train Hypro and ConML-Hypro are provided in Algorithm 10 and 11. The structure of $H$ is provided in Table 6, and two such hypernetworks are used for generate parameters for FiLM function. We implement ConML with $B = 16$, $\phi(a,b) = 1 - \frac{a\cdot b}{\|a\|\|b\|}$ (cosine distance) and $\lambda = 0.1$. As for the sampling strategy $\pi_\kappa$ and times $K$, for every task, we sample subset with different sizes, including each $m \in \{4, 8, 16, 32, 64\}$, for $128/m$ times respectively. A $m$-sized subset contains $m/2$ positive and $m/2$ negative samples sampled randomly. The other hyperparameters of model structure and training procedure follow the benchmark's default setting [40].

**Results.**  Table 7 shows the results. ConML-Hypro shows advantage over SOTA approach under all meta-testing scenarios with different shots. Comparing Hypro and ProtoNet, we can find the

**Algorithm 11** ConML-Hypro

---

**Note:** Refer to Algorithm 10 for details about $H_\theta(\mathcal{D})$ and $h_{\theta,\alpha}$.
**Input:** Task distribution $p(\tau)$, batch size $B$, inner-task sample times $K$ and sampling strategy $\pi_\kappa$.
**while** Not converged **do**
    Sample a batch of tasks $\boldsymbol{b} \sim p^B(\tau)$.
    **for** All $\tau \in \boldsymbol{b}$ **do**
        **for** $k = 1, 2, \cdots, K$ **do**
            Sample $\kappa_k$ from $\pi_\kappa(\mathcal{D}_\tau^{\text{tr}} \cup \mathcal{D}_\tau^{\text{val}})$;
            Get model representation $\boldsymbol{e}_\tau^{\kappa_k} = H_\theta(\kappa_k)$;
        **end for**
        Get model representation $\boldsymbol{e}_\tau^* = H_\theta(\mathcal{D}_\tau^{\text{tr}} \cup \mathcal{D}_\tau^{\text{val}})$;
        Get inner-task distance $D_\tau^{\text{in}}$ by (2);
        Get task-specific model $h_{\theta, H_\theta(\mathcal{D}_\tau^{\text{tr}})}$;
        Get validation loss $\mathcal{L}(\mathcal{D}_\tau^{\text{val}}; h_{\theta, H_\theta(\mathcal{D}_\tau^{\text{tr}})})$;
    **end for**
    Get $D^{\text{in}} = \frac{1}{B} \sum_{\tau \in \boldsymbol{b}} D_\tau^{\text{in}}$ and $D^{\text{out}}$ by (3);
    Get loss $L$ by (4);
    Update $\theta$ by $\theta \leftarrow \theta - \nabla_\theta L$.
**end while**

---

Table 6: Hypernetwork structure in Hypro and ConML-Hypro

|  | Layers | Output dimension |
|---|---|---|
| $\text{MLP}_1$ | input $[x_i \mid y_i]$ (dim=2562), fully connected, LeakyReLU | 2560 |
|  | $2\times$ fully connected with with residual skip connection | 2560 |
| $\text{MLP}_2$ | $2\times$ fully connected with residual skip connection, LeakyReLU | 2560 |

introduced hypernetwork can brings notable improvement. Comparing ConML-Hypro and Hypro, we can find the effect of ConML is significant.

Table 7: Few-shot molecular property prediction performance ($\Delta$AUPRC) on FS-Mol. † indicates result from [36]. ∗ indicates new approach proposed in this paper.

|  | 2-shot | 4-shot | 8-shot | 16-shot |
|---|---|---|---|---|
| MAML | $.009 \pm .006$ | $.125 \pm .009$ | $.146 \pm .007$ | $.159 \pm .009$ |
| PAR | $.124 \pm .007$ | $.140 \pm .005$ | $.149 \pm .009$ | $.164 \pm .008$ |
| ProtoNet | $.117 \pm .006$ | $.142 \pm .007$ | $.175 \pm .006$ | $.206 \pm .008$ |
| CNP | $.139 \pm .004$ | $.155 \pm .008$ | $.174 \pm .006$ | $.187 \pm .009$ |
| Hypro∗ | $.122 \pm .007$ | $.150 \pm .006$ | $.185 \pm .008$ | $.216 \pm .007$ |
| IterRefLSTM† | - | - | - | $.234 \pm .010$ |
| ADKF-IFT | $.131 \pm .007$ | $.166 \pm .005$ | $.202 \pm .006$ | $.234 \pm .009$ |
| ConML-Hypro∗ | $\mathbf{.175} \pm .006$ | $\mathbf{.196} \pm .006$ | $\mathbf{.218} \pm .005$ | $\mathbf{.239} \pm .007$ |

