# OpenReview forum: "Learning to Learn with Contrastive Meta-Objective"
_NeurIPS.cc/2024/Conference — Submitted to NeurIPS 2024_

### Official Review · Reviewer_eZHU · 2024-07-10

**Soundness:** 3
**Presentation:** 3
**Contribution:** 2
**Rating:** 4
**Confidence:** 4

**Summary:**

This study presents a contrastive regularizer to improve meta-learning. Specifically, the authors propose to incorporate a contrastive meta-objective that improves the alignment and the discrimination abilities of meta-learners, leading to better task adaptation and generalization. The authors demonstrate empirical effectiveness of the proposed ConML across several meta-learning and in-context learning scenarios.

**Strengths:**

1. The introduction of contrastive regularization sounds intuitively straightforward and motivating.
2. While equipping meta-learning with contrastive objective is not a new concept (e.g., [1], [2]) the implementation covers major meta-learning methods, including optimization- metric- and amortization-based methods. This means that the study is more comprehensive than previous studies.
3. The numerical results are promising. Code is provided - reproducibility is commendable.


[1] Gondal et al. Function Contrastive Learning of Transferable Meta-Representations. In ICML 2021.
[2] Mathieu et al. On Contrastive Representations of Stochastic Processes. In NeurIPS 2021.

**Weaknesses:**

My primary concern lies in the specific contrastive strategy employed.

1. The contrastive objective aims to minimize intra-task distances while maximizing inter-task distances. However, the absence of appropriate regularization or constraints raises concerns about potential model representation collapse. This collapse could manifest as representations converging to trivial solutions, such as constant vectors or confinement to low-dimensional subspaces. The authors should address whether they have considered these risks.
2. In addition, I wonder why the contrastive objective does not follow commonly studied ones, e.g., InfoNCE, in contrastive learning.
3. Moreover, the meta-objective necessitates computations involving representations from different tasks within a batch during each episode. Since the paper lacks a discussion on training and inference efficiency, the impact of this strategy on scalability is unclear.
4. The effectiveness of this method is likely dependent on hyperparameters tuning and the sampling strategy for creating subsets of tasks. As aforementioned, incorporating contrastive learning into meta-learning is not something completely new, even though, the authors do not include detailed discussion on how different strategies would affect the performance/efficiency, which is something to be expected on my end.

**Questions:**

please see weaknesses section. I am open to revise the score if reasonable clarifications can be provided.

**Limitations:**

The authors do have discussed the limitation of this study.

---

> ### Author Rebuttal · Authors · 2024-08-07
>
> Thank you for dedicating your time to give valuable comments on our paper. We especially find [1,2] are reasonable and interesting related works and we would like to discuss and credit to them in later paper.
>
> # Contrastive strategy, objective and hyper-parameters.
> **Reply.**
> Please refer to the *Common Reply 2*.
>
> # Training and inference efficiency.
> ConML is specially designed to be efficiently incorporated in the episodic training. As discussed at line 157~158, ConML introduces hardly no additional time-complexity, and the column "relative time" in Table 3,4 provides empirical verification. As for the memory complexity, we provide ablation study varying $K$. Please refer to the *Common Reply 2*. As for inference efficiency, ConML does not make any modification of the meta-testing procedure, thus as efficient as the original meta-learning algorithm. Thus ConML is feasible at scale.

---

### Official Review · Reviewer_rrDw · 2024-07-10

**Soundness:** 3
**Presentation:** 3
**Contribution:** 3
**Rating:** 6
**Confidence:** 5

**Summary:**

The paper aims to enhance the meta-learning process by implementing more robust supervision within the model space. Specifically, the authors seek to augment learning capabilities through model alignment and discrimination, aiming to approximate human-like rapid learning abilities. They propose that models trained on tasks within the same super-task exhibit similarity, while those trained on different task sets generalize effectively across diverse tasks. To achieve this, the authors devise a contrastive framework that remains independent of the specific meta-learning algorithms employed. This framework encourages closer alignment of representations for models adapted to similar tasks while pushing representations apart for models derived from dissimilar tasks. Moreover, the framework seamlessly integrates with optimization-based, metric-based, and amortization-based meta-learning methods. The approach demonstrates improvements across standard benchmarks in all evaluated scenarios. Furthermore, the proposed method shows promise for integration into the in-context learning of large language models, resulting in observed enhancements.

**Strengths:**

•  The paper is well-written and easy to understand. It's reasonable to enforce the model to emulate human learning capabilities through alignment and discrimination.

•  The proposed contrastive learning framework is versatile and applicable to most meta-learning methods.

•  The proposed method consistently improves upon existing meta-learning methods across standard benchmarks.

**Weaknesses:**

•  The paper lacks validation on MetaDataset[A], which is a common large-scale dataset for few-shot learning tasks.

•  There is a need for sensitivity analysis on certain hyperparameters, such as λ and the choice of similarity function for contrastive learning.

[A] Meta-Dataset: A Dataset of Datasets for Learning to Learn from Few Examples, ICLR2020.

**Questions:**

When sampling batches of tasks at each iteration, if the same class appears in both tasks, it raises questions about how discrimination is computed between tasks.

**Limitations:**

Please see weakness.

---

> ### Author Rebuttal · Authors · 2024-08-07
>
> Thank you for dedicating your time to give valuable comments on our paper.
>
> # The paper lacks validation on MetaDataset, which is a common large-scale dataset for few-shot learning tasks.
> **Reply.**
> As mentioned in *Common Reply 2*, currently only very few effort has been made to tune hyperparameter settings for performance, so we did not try ConML on large-scale dataset or SOTA methods. According to the reviewer's kind advices, here we provide ConML's empirical results on a common large-scale dataset, MetaDataset [1], and SOTA meta-learning method, CAML [2].
>
>
> Table 1 shows results on MetaDataset, experimented in the same model, training (on ILSVRC-2012 only) and evaluation setting following [1], and the ConML is introduced with the same setting as section 5.2. It can be observed that ConML also brings consistent improvement on MetaDataset.
>
> *Table 1. Few-shot image classification accuracy (%) on MetaDataset.*
> |Baseline| ILSVRC| Omniglot |Aircraft |Birds| Textures |Quick Draw| Fungi |VGG Flower| Traffic Signs |MSCOCO|
> |---- |  ----  |  ----  | ----  |----  | ----  |----  | ----  |----  |----  |----  |
> |MatchNet| 45.0|52.2|48.9|62.2|64.1|42.8|33.9|80.1|47.8|34.9|
> |ConML-MatchNet|  **51.1** |**54.6**|**51.5**| **66.8**| **67.6**|**46.7**| **36.4**| **84.9**| **49.5**|**40.1**|
> |
> |ProtoNet|  50.5| 59.9| 53.1| **68.7**| 66.5| 48.9| 39.7| 85.2| 47.1| 41.0|
> |ConML-ProtoNet|  **52.3**|**61.2**|**54.9**|**68.9**|**68.4**|**50.0**|**40.9**| **88.0**| **48.6**|**42.4**|
> |
> |fo-MAML|  45.5| 55.5| 56.2| 63.6| 68.0| 43.9| 32.1| 81.7| 50.9| 35.3|
> |ConML-fo-MAML|  **54.1** |**63.7**|**64.9**| **69.9**| **72.3**| **48.5**| **40.6**| **90.4**|**52.2**| **43.5**|
> |
> |fo-Proto-MAML| 49.5| 63.3| 55.9| **68.6**| 66.4| 51.5|39.9| 87.1| 48.8| 43.7|
> |ConML-fo-Proto-MAML|  **54.3** |**69.8**|**61.5**| **68.6**| **69.4**| **53.1**| **43.7**| **91.0**|**51.5**|**48.9**|
>
>
> We also study ConML's performance on SOTA meta-learning method Context-Aware Meta-Learning (CAML) [2]. CAML is based on in-context learning, composed of a feature extractor and an in-context learner (transformer). The feature extractor is pretrained and frozen during meta-training, which ConML does not effect. ConML effect on the training process of the in-context learner following the way introduced in section 4. We follow the same experiment settings in [2], that the feature extractor is a ViT-based model with initial parameters provided by CLIP [9]. The in-context learner is trained on ImageNet-1k, Fungi, MSCOCO, and WikiArt, note that this process is called 'large-scale pretraining' in [8], while it is regarded as the meta-training process in ConML. We introduce ConML with $K=1$, $\pi=$ *random half*, $\phi$ is cosine distance evaluated by the output of transformer before the readout MLP. The other training and evaluation settings stay the same following the public code from https://github.com/cfifty/CAML. The results are provided in Table 1, where all settings are 5-way (n)-shot. We find that ConML-CAML's performance is consistently better than CAML's, proving that ConML is an algorithm-agnostic approach that can bring universal improvement.
>
> *Table 2. Performance comparison of ConML-CAML and CAML (Accuracy %).*
> |Benchmark|  MiniImageNet,n=1 | n=5|  tiered-ImageNet,n=1 | n=5 | Aircraft,n=1  | n=5 | CIFAR-fs,n=1|n=5|
> |---- |  ----  |  ----  | ----  |----  | ----  |----  | ----  |----  |
> |CAML|  96.2 |98.6|95.4| **98.1**| 63.3| 79.1| 70.8| 85.5|
> |ConML-CAML| **97.0** | **98.9**|**96.6** |**98.2**|**65.8** | **81.5**| **72.3**| **86.1**|
>
> # There is a need for sensitivity analysis on certain hyperparameters, such as λ and the choice of similarity function for contrastive learning.
> **Reply.**
> Please refer to *Common Reply 2*.
>
> # When sampling batches of tasks at each iteration, if the same class appears in both tasks, it raises questions about how discrimination is computed between tasks.
> **Reply.**
> Yes, indeed. As we discussed in conclusion, as tasks are not likely to distribute equilaterally in the model representation space, batch-sampling strategy can certainly be detailed and improved. Currently ConML is statistically effective and left to be improved in many directions. In fact, the sample strategies can be viewed as a hyper-parameter setting about ConML, please refer to *Common Reply 2* for more information.
>
> [1] Triantafillou, E., Zhu, T., Dumoulin, V., Lamblin, P., Evci, U., Xu, K., ... & Larochelle, H. (2019). Meta-dataset: A dataset of datasets for learning to learn from few examples. arXiv preprint arXiv:1903.03096.
>
> [2] Fifty, C., Duan, D., Junkins, R. G., Amid, E., Leskovec, J., Ré, C., & Thrun, S. (2023). Context-aware meta-learning. arXiv preprint arXiv:2310.10971.
>
> [3] Radford, A., Kim, J. W., Hallacy, C., Ramesh, A., Goh, G., Agarwal, S., ... & Sutskever, I. (2021, July). Learning transferable visual models from natural language supervision. In International conference on machine learning (pp. 8748-8763). PMLR.

---

> > ### Comment · Reviewer_rrDw · 2024-08-12
> > **Response to Authors**
> >
> > I would like to thank the authors for their time and effort in addressing my concerns. My major concerns have been resolved and the evaluation of MetaDataset has shown great improvement. After reading other reviewers’ comments and authors’ rebuttals, I decide to maintain my score and remain positive.

---

> > > ### Author Response · Authors · 2024-08-13
> > > **Thanks**
> > >
> > > Thanks again for your response.

---

### Official Review · Reviewer_6XwB · 2024-07-13

**Soundness:** 2
**Presentation:** 2
**Contribution:** 2
**Rating:** 3
**Confidence:** 4

**Summary:**

This paper deals learning to learn (meta-learning) problem, from the perspective of exploring inner-task and intra-task relationship. Specifically, this paper proposed a Contrastive meta-objective by exploring intra- and inter-task distances and severed as an additional term for training objective (in addtion to classification loss).

Experiments are conducted on both conventional few-shot image classification and in-context learning settings. Common benchmarks are used to compare with simple few-shot methods such as MAML, ProtoNet, SCNAPs. For In-context learning, simple synthetic functions are used for comparison.

**Strengths:**

- The overall method makes sense. Designing intra- and inter-distance to explore the task-level contrastive information and further introduce this into to the meta-learning objective is reasonable. The diverse tasks naturally compose the contrastive pairs, useful for training.
- The proposed method is general and can be applied on top of different few-shot classification/regression methods, such as metric-based, optimisation-based, simpleCNAPs, and in-context learning.
- The performance gains over these simple baselines are significant, showing the effectiveness of the proposed method.

**Weaknesses:**

- Technically, the proposed contrastive meta-objective is similar to the idea of supervised contrastive learning, which already provides good insights to the representation learning and deep learning community. Therefore, the proposed method is kind of incremental and provides less new knowledge to the field.
- The method is only verified on top of simplest baseline methods (MAML, ProtoNet, etc). In meta-learning, various works have been proposed to investigate the possible exploration of the task-level information for improved meta-learning, such as [R1-R4], to name a few.
However, none of those previous efforts were discussed or compared. Only beating the naive baseline cannot comprehensively demonstrate the advantages of this paper.
- Experimentally, the proposed method tries to show its superior performance over simple baselines rather than SOTA. This is less convincing.
- The in-context learning experiments are only on simple synthetic data, lack of significance.
- The tile and scope: Learning-to-learn is very general, but in fact only classification related experiments are conducted. By convention, the learning-to-learn approaches will also verify on reinforcement learning.

[R1] Fei, N., Lu, Z., Xiang, T., & Huang, S. (2021). MELR: Meta-learning via modeling episode-level relationships for few-shot learning. In International Conference on Learning Representations.
[R2] Agarwal, P., & Singh, S. (2023). Exploring intra-task relations to improve meta-learning algorithms. arXiv preprint arXiv:2312.16612.
[R3] Han, J., Cheng, B., & Lu, W. (2021). Exploring task difficulty for few-shot relation extraction. arXiv preprint arXiv:2109.05473.
[R4] Zhang, Tao. "Episodic-free Task Selection for Few-shot Learning." arXiv preprint arXiv:2402.00092 (2024).

**Questions:**

Please see above Weaknesses

---

> ### Author Rebuttal · Authors · 2024-08-07
>
> Thank you for dedicating your time to give valuable comments and advises on our paper.
>
> # Distinguishing ConML with conventional contrastive learning.
>
> Conventional contrastive learning methods have already provided good insights to the representation learning and deep learning community. It has been successfully used to learn representations or encoders in both supervised and unsupervised learning. However, what can this insight brings to, and how can it perform successfully in meta-learning, have not been explored before. This paper brings the insight and practice of contrastive learning from (un)supervised learning to meta-learning with providing new insight and a practical and universal framework.
>
> On the other hand, ConML does not seek for technical contribution to contrastive learning methods. How to improve meta-leaner's ability of fast-adaptation and generalization to diverse tasks is an essential problem for meta-learning, and ConML is a universal, effective and efficient solution. Inspired by human's learning capacity, ConML encourages meta-learners to perform alignment and discrimination, which is original. This is achieved by minimizing inner-task distance and maximizing inter-task distance respectively. For efficient and universal integration with the general episodic meta-training, ConML is intuitively designed with such a "sample and contrast" implementation. Please refer to *Common Reply 1* for the primary contribution.
>
> # Comparing ConML with previous efforts to investigate the possible exploration of the task-level information for improved meta-learning.
>
> Thanks a lot for your kind nomination of related works, and we would like to add to the paper to discuss and credit exploration of the task-level information for improving meta-learning.
> However, ConML is not comparable with existing efforts, but can be adopted beyond those methods, due to its algorithm-agnostic and versatile property.
> Focusing on the most general meta-learning setting on top of different meta-learning models and algorithms, as illustrated in section 3.0, ConML introduces an algorithm-agnostic objective to improve the episodic meta-training process. Thus
> ConML can be generally adopted to improve existing meta-learning algorithms.
> For [1,5,6] and [paper-ref.51,52,55], they constrain on a few-shot classification setting where there is a pool of base classes for meta-training where each class contains abundant data. Though they design different meta-objectives for each specific setting, ConML can be generally incorporated by replacing the naive validation loss in eqn.(4) with their loss. There are also works investigating like meta-training with curriculum [2,3,5] and meta-batch-diversity [7], thus resulting in certain batch sampling strategies in the episodic training. ConML can also be incorporated with their specific sampling strategies as discussed in conclusion.
>
> # Integration ConML with SOTA meta-learning models
>
> As illustrated above, ConML focuses on the most general meta-learning setting on top of different meta-learning models and algorithms. ConML is algorithm-agnostic after defining the model representation $\psi(g())$. As the design of $\psi(g())$ is simple and intuitive, we show specification for representative meta-learning methods for above four basic categories. And experiments in Table 3,4 and 5 show the effectiveness for every category. As SOTA meta-learning methods can be viewed as enjoying a combination of the these basic categories, it is easy to integration ConML by specifying $\psi(g())$ for the method, without loss of generalizability of the effect.
>
> Here we study ConML's performance on SOTA meta-learning method Context-Aware Meta-Learning (CAML) [8]. CAML is based on in-context learning, composed of a feature extractor and an in-context learner (transformer). The feature extractor is pretrained and frozen during meta-training, which ConML does not effect. ConML effect on the training process of the in-context learner following the way introduced in section 4. We follow the same experiment settings in [8]. The training of the in-context learner is called 'large-scale pretraining' in [8], while it is regarded as the meta-training process in ConML. We introduce ConML with $K=1$, $\pi=$ *random half*, $\phi$ is cosine distance evaluated by the output of transformer before the readout MLP. The other training and evaluation settings stay the same following the public code from https://github.com/cfifty/CAML. The results are provided in Table 1, where all settings are 5-way (n)-shot. We find that ConML-CAML's performance is consistently better than CAML's, proving that ConML is an algorithm-agnostic approach that can bring universal improvement.
>
> *Table 1. Performance comparison of ConML-CAML and CAML (Accuracy %).*
> |Benchmark|  MiniImageNet,n=1 | n=5|  tiered-ImageNet,n=1 | n=5 | Aircraft,n=1  | n=5 | CIFAR-fs,n=1|n=5|
> |---- |  ----  |  ----  | ----  |----  | ----  |----  | ----  |----  |
> |CAML|  96.2 |98.6|95.4| **98.1**| 63.3| 79.1| 70.8| 85.5|
> |ConML-CAML| **97.0** | **98.9**|**96.6** |**98.2**|**65.8** | **81.5**| **72.3**| **86.1**|
>
> # The in-context learning experiments are only on simple synthetic data, lack of significance.
>
> We study the effect of ConML on in-context learning on real data through, ConML-CAML, provided in above Table 1. We apologize for not being able to study on conventional NLP with training a LLM due to the limit of resources.
>
> [1] MELR: Meta-learning via modeling episode-level relationships for few-shot learning
>
> [2] Exploring intra-task relations to improve meta-learning algorithms.
>
> [3]  Exploring task difficulty for few-shot relation extraction
>
> [4] Episodic-free Task Selection for Few-shot Learning
>
> [5] Progressive meta-learning with curriculum
>
> [6] Adaptive task sampling for meta-learning
>
> [7] The effect of diversity in meta-learning
>
> [8] Context-aware meta-learning
>
> [9] Learning transferable visual models from natural language supervision

---

### Official Review · Reviewer_KosZ · 2024-07-13

**Soundness:** 2
**Presentation:** 1
**Contribution:** 2
**Rating:** 4
**Confidence:** 4

**Summary:**

The paper proposes a contrastive meta-objective that can be applied to various meta-learning methods. Also, interpreting in-context learning as a meta-learning formulation, extended the proposed method to in-context learning. Specifically, the objective is to contrast task identity obtained after episode optimization. The task identity is defined as the model weight or the feature obtained by feed-forwarding, in the case of in-context learning. Finally, the authors demonstrated the superiority of the proposed method by applying it to several meta-learning methods to improve their performance.

**Strengths:**

- The proposed method can improve diverse meta-learning methods.
- Contrasting task identity sounds intuitive.

**Weaknesses:**

In general, the text is not easy to understand.

- Not self-stained figures.
  - In Figure 1, it's hard to understand what $h_{w_i}$ and $w_i$ are since the caption has no explanation.
  - In Table 1, the caption could have included the definition of $g$ or $\psi$.
  - Figure 2 is hard to read; (b) and (e) missed the x, y-axis meaning and are too small to see something.
- Experiment details are missing.
  - Hard to understand Section 5.1. Though the sine wave regression problem is well-known in this domain, it's hard to interpret results without task definitions.
  - The tasks in Section 5.3 are not clearly defined.
- An ablation study would be helpful.
  - How to decide distance function $\phi$?
  - What if increasing the number of task-sampling $K$?

**Questions:**

- In the case of optimization-based methods, should the method need to proceed the episode optimization twice, once for $D^{tr}\cup D^{val}$ and once for $D^{tr}$? Then why does it only take 1.1~1.5x time only?
- In the case of metric-based methods, if class order is shuffled, the model presentation $[c_1|c_2|...|c_N]$ also changes?
- In the case of ICL, does "consistent value in an episode" mean the same value is used for every task and contrast
  - Can using a feature after feed-forwarding a random input to the model be also used for the other settings, i.e. meta-learning methods?
- In case of two tasks share almost the same classes (e.g. task 1 consists of class 1,2,3 and task 2 consists of class 1,2,4), should we consider these two tasks to be discriminated much?
- Why "alignment" is related to fast-adaptation? Isn't fast adaptation related to the number of steps to converge, not the number of shots?
- Why "discrimination" is related to the generalization?

---

> ### Author Rebuttal · Authors · 2024-08-07
>
> Thank you for dedicating your time to give valuable comments on our paper, especially for valuable advises about improving the presentation.
> # In the case of optimization-based methods, should the method need to proceed the episode optimization twice? Then why does it only take 1.1~1.5x time only?
> **Reply.**
> To directly answer 'why does it only take 1.1~1.5x time only?': Though ConML introduces additional inner-update processes using different subsets, the processes are independent with each other thus can be parallel computed.
>
> ConML does not introduce additional episode optimization procedure integrating with any meta-learning methods. We apologize for possibly leading to misunderstanding and want to re-interpret the proposed ConML here. Different meta-learning methods are unified and formulated in an episodic training manner. In each episode, the meta-learner function $g$ generates model $h$ with $\mathcal{D}^{tr}$, i.e., making adaptation according to $\mathcal{D}^{\text{tr}}$, then episode optimization is performed to minimize the validation loss of $h$ on $\mathcal{D}^{\text{val}}$ (eqn.(1)). ConML requires multiple subset samples from $\mathcal{D_{\tau}}^{\text{tr}}\cup\mathcal{D_{\tau}}^{\text{val}}$ to be processed independently by $g$, which are parallel computed inside an episode, to obtain the contrastive objective (eqn.(2)(3)). Then to perform the episode optimization, the contrastive objective and validation loss are added up (eqn.(4)), and minimized by gradient descent. The specifications of meta-learner function $g$ are provided in Table 1.
> In the case of optimization-based methods, taking ConML-MAML for example (Algorithm 6), the meta-learner function $g$ corresponds to the inner-update. ConML only requires additional inner-update subsets of $\mathcal{D_\tau}^{\text{tr}}\cup\mathcal{D_\tau}^{\text{val}}$, which are independent and parallel computed. Then ConML only need to proceed the episode optimization (global-update) once. Thus ConML introduces no additional time complexity, as illustrated at line 157.
>
> # In the case of metric-based methods, if class order is shuffled, the model presentation also changes?
> **Reply.**
> Yes. Metric-based methods can be viewed as equipped with classifiers with class-order-aware weights.
> We represent the model with the weights which are class-order-aware. Taking ProtoNet for example, the classifier can be viewed as a linear model with order-aware weights (eqn.(8) in [paper-ref 39]). If class order is shuffled, we do need different model to make prediction, thus the the model presentation also changes.
>
> # In the case of ICL, does "consistent value in an episode" mean the same value is used for every task and contrast? Can using a feature after feed-forwarding a random input to the model be also used for the other settings, i.e. meta-learning methods?
> **Reply.**
> The $u$ for every of the $B$ tasks in a certain episode, is necessary to be consistent to calculate inter-task distance. However, for tasks not in the same episode (batch), no contrastive objective among them would be evaluated. Thus $u$ in different episode can be different, while necessarily being consistent in an episode.
> Technically, obtaining model representation by input probing can be used for other meta-learning methods. However, we do not recommend this. Because it is less efficient and less expressive than model representation by explicit weights. It is less efficient because requiring additional feed-forwarding. It is less efficient because the difficult to cover the input space. We use explicit weight for other meta-learning methods while input probing for ICL, because we could not obtain task-specific explicit weights of ICL model.
>
> # In case of two tasks share almost the same classes (e.g. task 1 consists of class 1,2,3 and task 2 consists of class 1,2,4), should we consider these two tasks to be discriminated much?
> **Reply.**
> Yes, Good idea. As we discussed in conclusion, as tasks are not likely to distribute equilaterally in the model representation space, batch-sampling strategy can certainly be detailed and improved. Currently ConML is statistically effective and left to be improved in many directions. In fact, the sample strategies can be viewed as a hyper-parameter setting about ConML, please refer to *Common Reply 2* for more information.

---

> ### Author Response · Authors · 2024-08-07
> **Complement Rebuttal**
>
> *Sorry for the split of our rebuttal due to the character limit.*
> # Why "alignment" is related to fast-adaptation? Isn't fast adaptation related to the number of steps to converge, not the number of shots?
> **Reply.**
> From the perspective of gradient-based meta-learning methods, "fast" can be interpreted as "few steps" to converge. From a more general perspective of meta-learning, "fast" could be interpreted as "few-shot", for the following reasons: In real world application of machine learning, the accumulation of labeled data takes time; Apart from gradient-based methods, other meta-learning methods do not have the concept of "steps" for adaptation;  Early meta-learning works take shots as sequential input (e.g., with RNN, memorization), thus temporal fast means short sequence, corresponding to few-shot.
>
> "Alignment" is mapping the representations of a positive pair together in contrastive learning [paper-ref 48], thus be (mostly) invariant to unneeded noise factors. In ConML, this means aligning model generated by the meta-leaner with different subsets of the same task (line 140~143), by minimizing $D^{in}$ (eqn.(2)). This encourage the model generated with few-shot to be similar with fully-supervised ones. The meta-learner thus can learn noise- and bias-invariant task knowledge from few-shot.
> Experiment results in Fig.2(c) verifies "alignment" is related to fast-adaptation.
>
> # Why "discrimination" is related to the generalization?
> **Reply.**
> "Discrimination" is maximizing intra-task distance, i.e., distance between negative pairs in contrastive learning (line).
> Meta-learning aims at improving the performance on unseen tasks. With a natural supposition that different tasks enjoy different task-specific models, discrimination, i.e., meta-learner can learn different
> models from different tasks, is important for task-level generalizability.
> Experiment results in Fig.2(f) verifies "discrimination" is related to task-level generalization.
> This objective also plays the role as an offset of the inner-task distance. In fact, this objective can be not optimized explicitly but through serving as negative pairs to optimize the softmax value in InfoNCE in *Common Reply 2* and obtain better performance.
>
> # Illustration about experiment results and details.
> **Reply.**
> Due to the page limit, details about experiment settings have been presented as short as possible (e.g., "following the same settings in [17]", "Following [18]"). We apologize for the difficulty in reading. However, we hope the omission of unmentioned details would not trouble the understanding of the experiment results, and code for all experiments is provided in appendix. We would like to provide full details in appendix later.
>
> As for section 5.1, we provide analysis about statistical results comparing meta-training MAML with different meta-objectives (MAML, in-MAML, out-MAML, ConML-MAML).
> Result 1: Comparing Fig.2(a) and (d), which are t-sne visualization of model representations, along with the clustering performance in Table 2, ConML do align and discriminate the model representation as expected.
> Result 2: Fig.2(b) and (e) (x-axis: cosine distance value, y-axis: count) show the distribution of inner-task and inter-task distance respectively, meta-trained by the four variants.  We can find that: the alignment and discrimination ability corresponds to optimizing inner- and inter-task distance respectively; the alignment and discrimination capabilities are generalizable; ConML shows the couple of both capabilities.
> Result 3: Fig.2(c) shows the testing performance varying shots number (x-axis: shot, y-axis: Relative Error (/MAML)). We find alignment is closely related to the fast-adaptation ability of the meta-learner.
> Result 4: Fig.2(f) shows the testing performance varying the distribution of sine parameter (x-axis: Distribution Shift, y-axis: Relative Error (/MAML)). We find discrimination is closely related to the task-level generalizability.
>
> # Ablation studies.
> **Reply.**
> Please refer to *Common Reply 2*.

---

### Author Rebuttal · Authors · 2024-08-06

Dear all reviewers,

Thank you for dedicating your time to give valuable comments on our paper!
Here we want to response globally for the primary contribution, and experiments about some detail settings about ConML.

# Common Reply 1:
## Primary contribution.
We regard the primary contribution of ConML as providing the universal and efficient framework on top of the most general meta-learning setting and training procedure, to encourage meta-leaner's alignment and discrimination ability exploiting task identity. The proposed framework can be integrated in any existing meta-learning models and algorithms, i.e., algorithm-agnostic. Currently the implementation of ConML is somewhat primitive, but it lays the groundwork and can be easily tailed and improved in many directions.

# Common Reply 2:
## Hyper-parameter settings and ablation studies.
As we discussed in conclusion, in the current paper ConML follows the simple and intuitive design of batch sampling, subset sampling and contrastive strategies.
The experiment results shows that meta-training to align and discriminate following the ConML framework is very effective even though the strategies are primitive. In the current paper we use cosine distance, as it is a very simple choice with bounded value. And the choice of the naive contrastive loss is to decoupling inner- and inter-task distance to analyze the contribution of alignment and discrimination (sec 5.1). In fact, as mentioned at line 294, we have very simple and intuitive setting of ConML sharing with all baselines in experiment, which indicates ConML brings significant improvement without much hyper-parameter tuning.
Fully fulfilling ConML's potential by tuning the hyper-parameters is an interesting but difficult challenge due to the high dimension of hyper-parameters related to ConML's setting (e.g., contrastive loss and distance function, $\lambda$, $K$, $\pi$, $B$, w.r.t. each specific meta-learning algorithm).

In this reply we consider varying the setting of three important hyper-parameters: 1. subset-sampling number $K$, which determines the complexity; 2. distance function $\phi$.  3. Contrastive strategy, including $\lambda$, which weights the contrastive term in meta-objective, and Using InfoNCE loss, a widely known contrastive loss. 2 and 3 are studied coupling together. All results are obtained on miniImageNet 5-way 1-shot setting.

## 2.1 Ablation study about $K$.
From Table 1, we find that start from $K=1$, the performance shows a moderate growth with $K$. And the memory growth linearly with $K$. Note that between w/o ConML and $K=1$, there is discrepancy in both performance and memory (~2x). Note that $K$ does not have effect on time efficiency with enough memory space.

*Table 1. Ablation study about $K$*
|ConML- |   | w/o|     $K$= 1  | 2 | 4  | 8 | 16| 32|
|---- |  ----  |  ----  | ----  |----  | ----  |----  | ----  |----  |
| MAML| Acc. (%)  |48.75|56.25| 56.08| **57.59**| 57.40| 57.43| 57.33|
| MAML| Mem. (MB)  | 1331|2801 |2845 | 3011| 3383| 4103| 5531|
| ProtoNet| Acc. (%)  | 48.90|51.03| 51.46| 52.04| 52.30| 52.34| **52.48**|
| ProtoNet| Mem. (MB)  | 7955|14167|14563| 15175| 16757|19943 |26449|



## 2.2 Ablation study about distance function and contrastive strategy.
In ConML, the meta-objective to minimize is is $L=L_v+\lambda L_c$, where $L_v$ is the objective of the original meta algorithm, e.g., validation loss. The meta-objective requires $L_c$ to have lower bounded to optimize.

In the current paper, we have the naive contrastive loss $L_c=D^{in}-D^{out}$, and to be bounded we choose nature cosine distance $\phi(x,y)=1-\frac{x^{\top} y}{\Vert x\Vert\Vert y\Vert}$. A less nature choice is manually bounded euclidean distance $\phi(x,y)=sigmoid(\Vert x-y\Vert)$. Apart from the naive contrastive loss, here
we consider InfoNCE loss in an episode given a batch $b$ containing $B$ tasks by
$$L_c=-\sum_{\tau\in {b}}log(\frac{exp(-D_{\tau}^{in})}{exp(-D_{\tau}^{in})+\sum_{\tau'\in b/\tau}exp(-D_{\tau,\tau'}^{out})}),$$
where $D_{\tau,\tau'}^{out}=\phi({e}^*_{\tau},{e}^*_{\tau'})$. Note that we regard negative "distance" as "similarity". We consider cosine distance $\phi(x,y)=1-\frac{x^{\top} y}{\Vert x\Vert\Vert y\Vert}$ and euclidean distance $\phi(x,y)=\Vert x-y\Vert$.

Table 2 shows the results. We find that ConML in a considerable range of $\lambda$ can bring meta-learners significant performance improvement, while a too large $\lambda$ would result in model collapse due to diluting meta-leaner's original objective. Different algorithms prefer different choice of distance function. InfoNCE shows superiority over the naive contrastive strategy, with higher potential and much less sensitivity. This indicates that we might have not been close to the best ConML can do, and it can be easily improved in many directions.

*Table 2. Ablation study about contrastive strategy and distance function.*
|ConML- |  $L_c$ |$\phi$ |     $\lambda$=0| 0.01  | 0.03 | 0.1 | 0.3 | 1|
|---- |  ----  |  ----  | ----  |----  | ----  |----  | ----  |----  |
| MAML| naive |cosine|48.75|52.19 | 54.43| 56.25| 55.82| 47.39|
| MAML| naive  | sigmoid(euc)| 48.75| 51.64| 54.40| 54.06| 53.94|54.21 |
| MAML| InfoNCE  | cosine|48.75| 54.66| 55.90|**57.24** | 56.87| 56.95|
| MAML| InfoNCE  | euc|48.75|53.02| 55.08| 55.61| 55.89|55.40|
|
| ProtoNet| naive |cosine|48.90| 49.16|51.58 | 51.03| 50.06| 48.81|
| ProtoNet| naive  | sigmoid(euc)|48.90 |50.27|51.45|52.09 |52.80 | 52.02|
| ProtoNet| InfoNCE  | cosine|48.90| 50.73| 52.20| 52.44| 52.86| 52.15|
| ProtoNet| InfoNCE  | euc|48.90|51.54| 52.39| 53.42| 53.30|**53.81**|

---

### Author Response · Authors · 2024-08-12
**Gentle Reminder**

Dear all reviewers,

We appreciate your sincere and constructive feedback on our paper. During the rebuttal period, we think our paper was improved to answer all your concerns and questions.

We also believe that our paper can be further advanced during this discussion period. We would appreciate it a lot if reviewers could discuss the response and re-evaluate our paper based on those discussions, and we would cherish any further comments and suggestions.

Sincerely,

Authors

---

### Public Comment · ~Shiguang_Wu2 · 2025-11-14
**New version has been accepted as NeurIPS 2025 (oral).**

https://arxiv.org/abs/2410.05975

---

### Decision · Program_Chairs · 2024-09-25

**Decision:**

Reject

**Comment:**

The paper proposes a generic contrastive meta-learning objective function compatible with standard episodic meta-learning one. The contrastive objective leverages task identity for additional supervision signals, minimizing model representations on tasks with same IDs, and maximizing tasks with different IDs. The objective also works with in-context learning. Experiments show notable improvements from several meta-learning baselines.

All reviewers indicated high confidence for their decisions and most didn't recommend acceptance. A key weakness highlighted by reviewers is the lack of more performant baselines, on which the new objectives could be tested. While the authors provided comparison for CAML, a more recent method, the improvements obtained there is much smaller compared to those presented in the main paper. In addition, I wish to highlight that the work does not discuss/compare to meta-learning methods that leverage pre-training, which achieves SOTA performance and compares outperforms the proposed method. Pre-training could be interpreted as a more fine-grained "alignment": individual samples from tasks must be matched to the appropriate classes [1].

Another concern raised by the reviewers is impact of the new objective function on training stability, which is not sufficiently addressed.

[1] Robust Meta-Representation Learning via Global Label Inference and Classification. (Wang et al., 2024)